# Mothers in a cooperatively breeding bird increase investment per offspring at the pre-natal stage when they will have more help with post-natal care

Pablo Capilla-Lasheras[1,2]*, Alastair J. Wilson[1], Andrew J. Young[1]*

**1** Centre for Ecology and Conservation, University of Exeter, Penryn, United Kingdom, **2** Current address: School of Biodiversity, One Health and Veterinary Medicine, University of Glasgow, Glasgow, United Kingdom

* pacapilla@gmail.com (PC-L); A.J.Young@exeter.ac.uk (AJY)

**Data Availability Statement:** All R scripts and datasets needed to reproduce the analyses presented in this paper are available at https://doi.org/10.5281/zenodo.8385995.

## Abstract

In many cooperative societies, including our own, helpers assist with the post-natal care of breeders' young and may thereby benefit the post-natal development of offspring. Here, we present evidence of a novel mechanism by which such post-natal helping could also have beneficial effects on pre-natal development: By lightening post-natal maternal workloads, helpers may allow mothers to increase their pre-natal investment per offspring. We present the findings of a decade-long study of cooperatively breeding white-browed sparrow-weaver, *Plocepasser mahali*, societies. Within each social group, reproduction is monopolized by a dominant breeding pair, and non-breeding helpers assist with nestling feeding. Using a within-mother reaction norm approach to formally identify maternal plasticity, we demonstrate that when mothers have more female helpers, they decrease their own post-natal investment per offspring (feed their nestlings at lower rates) but increase their pre-natal investment per offspring (lay larger eggs, which yield heavier hatchlings). That these plastic maternal responses are predicted by female helper number, and not male helper number, implicates the availability of post-natal helping per se as the likely driver (rather than correlated effects of group size), because female helpers feed nestlings at substantially higher rates than males. We term this novel maternal strategy "maternal front-loading" and hypothesize that the expected availability of post-natal help either allows or incentivizes helped mothers to focus maternal investment on the pre-natal phase, to which helpers cannot contribute directly. The potential for post-natal helping to promote pre-natal development further complicates attempts to identify and quantify the fitness consequences of helping.

## Introduction

Maternal effects arising from variation in pre-natal maternal investment in the egg or fetus can have profound fitness consequences for mothers and offspring [1–3]. In social organisms, mothers are predicted to evolve investment strategies that maximize their fitness returns on

**Funding:** The long-term field study was funded by a BBSRC David Phillips Research Fellowship to A.J. Y. (BB/H022716/1) and P.C.-L. was supported by a BBSRC-funded PhD studentship (BB/M009122/1). The funders had no role in study design, data collection and analysis, decision to publish, or preparation of the manuscript.

**Competing interests:** The authors have declared that no competing interests exist.

investment according to their social environment [4–6]. Cooperatively breeding species are of particular interest in this regard, as helpers typically contribute to the post-natal feeding of the offspring of breeding females (hereafter "mothers") and thus have the potential to impact the optimal level of maternal pre-natal investment per offspring [4,6–8]. Where mothers are assisted by variable numbers of helpers throughout their lives, selection may be expected to favor plastic strategies in which mothers adjust their pre-natal investment per offspring according to the likely availability of help during the post-natal period [4].

Different maternal strategies for adjusting pre-natal investment per offspring to the presence of helpers are hypothesized to evolve depending on how helpers impact the maternal payoff from pre-natal investment per offspring. The leading hypotheses (outlined below) focus on the mechanisms by which helpers could affect the optimal level of pre-natal investment per offspring, independent of variation in offspring number per breeding attempt (which could itself affect pre-natal investment per offspring via resource allocation trade-offs [9]). Empirical tests of these hypotheses therefore allow for the possibility that pre-natal investment per offspring is also affected by trade-offs with offspring number (e.g., clutch size in birds, which could itself be adjusted to helper number [10]).

Helpers have the potential to decrease the mother's optimal level of pre-natal investment per offspring, leading to strategies in which mothers reduce pre-natal investment per offspring when assisted by more helpers [4,7,11,12]. Such a maternal reduction in pre-natal investment per offspring when helped is typically referred to as a "load-lightening" response ([7,10,13]; following the historical use of this term to describe helper-induced reductions in maternal investment per offspring at the post-natal stage [5,14]). The "Load-Lightening Hypothesis" [5,7] for the adjustment of maternal pre-natal investment per offspring envisages that selection could favor such a maternal strategy if the contributions of helpers (i) increase the overall provision of post-natal care per offspring (a scenario that we term the provision of "additive" post-natal care by helpers following [5,15]) and thereby (ii) compensate, in part or whole, for any maternal reduction in pre-natal investment per offspring when helped (formally modeled as the "head start" scenario in [6]). Notably, this hypothesis requires that helper-derived post-natal care can compensate for reductions in maternal pre-natal investment (i.e., that investment can be "substituted across stages"; [6]), which may not always be the case [6,16,17]. Indeed, there is ample evidence that pre-natal conditions, and pre-natal maternal investment in particular, can have formative effects on offspring phenotype and performance [2,3,16–20].

Helpers also have the potential to increase the mother's optimal level of pre-natal investment per offspring, leading to strategies in which mothers instead increase pre-natal investment per offspring when assisted by more helpers [4,6,8,21,22]. The "Differential allocation hypothesis," for example, proposes that mothers should increase maternal investment under circumstances that increase their expected return on investment in their current breeding attempt, such as the presence of a high-quality mate or more helpers [4,8,21,23–26]. This hypothesis was originally proposed in the context of non-cooperative species [23,24,26], before being verbally extrapolated to cooperative breeders, with the suggestion that, as helpers commonly increase the reproductive value of offspring by providing "additive" post-natal care (i.e., increasing the overall provision of post-natal care per offspring [5,15]), mothers should increase investment per offspring when helped [4,8,13,21,25,27]. More specifically, the provision of additive post-natal care by helpers may increase the mother's return on pre-natal investment per offspring wherever pre- and post-natal investment have positive interactive effects on offspring fitness (such that post-natal helping increases the effect of maternal pre-natal investment on offspring fitness; [6]). Indeed, mathematical models incorporating such interactive effects of pre- and post-natal investment per offspring (the "silver spoon" scenario

in [6]) predict that, where helpers contribute to post-natal care, mothers should increase both pre- and post-natal investment per offspring when helped.

Cooperatively breeding birds provide a fruitful testing ground for these hypotheses, given the ability to estimate maternal pre-natal investment per offspring across different helping contexts by measuring egg traits. Several studies of cooperative birds have now reported that, after controlling for variation in clutch size, mothers with (more) helpers lay smaller eggs; the pattern predicted by the load-lightening hypothesis (e.g., *Malurus cyaneus* [7]; *Corvus corone corone* [11]; *Vanellus chilensis* [28]; *Philetairus socius* [12]; see also [29] for an experimental demonstration in fish). Three studies of cooperative birds have reported no evident relationship between egg size and the availability of help [10,30,31], and just 2 studies have reported the reverse relationship. Iberian magpie (*Cyanopica cooki*) and placid greenbul (*Phyllastrephus placidus*) mothers with more helpers lay larger eggs, consistent with the pre-natal predictions of the differential allocation hypothesis [6,21,22,25]. The situation may be more complex in some cases, however, as recent work suggests that the previously reported negative relationship between egg size and the availability of help in super fairywrens [7] becomes more positive under warmer conditions [8]. Given the overall weight of evidence for negative relationships across species, a meta-analysis of these collated findings has led to the suggestion that helpers commonly decrease the mother's optimal level of pre-natal investment per offspring and that the rationale of the load-lightening hypothesis may therefore commonly apply [13].

Crucially though, it has yet to be demonstrated that any of these associations between helper number and egg size in cooperative birds arise specifically from maternal plasticity (i.e., within-mother variation in egg size; see [29]). They could arise instead from among-mother variation in egg size being correlated with among-mother variation in helper number (e.g., mothers on higher-quality territories might simply lay larger eggs and have more offspring that survive to become helpers). Indeed, a study that explicitly teased apart the effects of within- and among-mother variation in helper number found that the negative relationship initially detected between helper number and egg volume in red-winged fairywrens (*Malurus elegans*) arose from among-mother variation in egg volume rather than maternal plasticity (i.e., within-mother variation; maternal plasticity was instead detected in clutch size), illustrating the importance of taking this approach [10]. While this same approach has revealed maternal plasticity in egg size according to abiotic conditions (e.g., temperature; [8]), evidence of maternal plasticity in egg size according to the availability of help per se does not yet exist for cooperative birds [10,31]. As such, it remains unclear whether avian mothers ever do adjust their pre-natal investment per offspring according to helper number, and whether any such maternal plasticity conforms to the predictions of the load-lightening or differential allocation hypotheses.

Here, we use a long-term field study of cooperatively breeding white-browed sparrow-weavers, *Plocepasser mahali*, to test the key predictions of these load-lightening and differential allocation hypotheses for the evolution of maternal plasticity in pre-natal investment. We do so by testing for maternal plasticity in both pre-natal investment per offspring (egg volume, while accounting for effects of clutch size) and post-natal investment per offspring (maternal nestling provisioning rate, while accounting for brood size) to the availability of help. We test for plasticity using a maternal reaction norm approach, in which we isolate the effects of within-mother variation in helper number on maternal investment (i.e., maternal plasticity) from potentially confounding effects of variation among mothers [10,31,32]. White-browed sparrow-weavers live in social groups of 2 to 12 birds, in which a single dominant female ("the mother") and male monopolize within-group reproduction and non-breeding subordinate "helpers" of both sexes help to feed their nestlings [33,34]. Helpers are typically past offspring of the dominant breeding pair and hence are usually helping to rear close kin [33]. Female

helpers feed nestlings at approximately twice the rate of male helpers, and, accordingly, female helper number has a demonstrably causal positive effect on the total rate at which broods are fed while male helper number does not (i.e., only female helpers provide demonstrably "additive" post-natal care; Fig 1 in [35]). That only female helpers provide demonstrably additive post-natal care provides an unusual opportunity to distinguish the hypothesized pre-natal maternal responses to the availability of additive help (which should manifest in this species as maternal responses to the number of female helpers) from maternal responses to group size more generally (which could influence maternal investment through mechanisms other than helping; [36,37]).

Sparrow-weaver mothers lay small clutches of 1 to 3 eggs (modal clutch size = 2 eggs) and do not adjust their clutch size according to helper numbers (see Results). Indeed, given their small clutch size, subtle adjustments in pre-natal maternal investment may be more readily achieved through changes in investment per egg than through changes in clutch size. The focal hypotheses assume that laying mothers are able to predict the helper numbers that they will have during the post-natal rearing period, in order to adjust their own pre-natal investment per offspring accordingly. This should be straightforward in sparrow-weaver societies, as both male and female helper numbers at laying strongly predict male and female helper numbers, respectively, during the post-natal rearing period (S1 Fig). We assess pre-natal maternal investment per offspring by quantifying egg volume, which, in this species, is strongly correlated with egg mass at laying and strongly predicts nestling mass at hatching (see Results). Maternal variation in egg volume is therefore likely to have fitness implications for offspring (and their mothers), not least because nestling mass at hatching positively predicts nestling survival to fledging in this species [35].

We test the following key predictions of the 2 focal hypotheses. The load-lightening hypothesis ("head start" scenario in [6]) predicts that sparrow-weaver mothers should decrease egg volume when assisted by more female, but not male, helpers. The differential allocation hypothesis ("silver spoon" scenario in [6]) predicts that sparrow-weaver mothers should increase both egg volume and their nestling provisioning rate when assisted by more female, but not male, helpers. To test these predictions, we first investigate whether within-mother variation in female and male helper numbers at laying predicts variation in egg volume (utilizing a large longitudinal data set; 490 eggs laid in 271 clutches by 62 mothers in 37 social groups; 1 to 21 eggs [median = 7] per mother). These analyses of egg volume control for any effects of variation in clutch size and allow for the possibility of interactions between helper numbers and clutch size (as, for example, the extent to which helper contributions can compensate for any reduction in pre-natal maternal investment when helped [under the load-lightening hypothesis] may depend upon the number of offspring in the brood). We also confirm that our findings are not complicated by parallel maternal plasticity in clutch size according to helper numbers, by verifying that within-mother variation in female and male helper numbers does not predict clutch size. As egg traits often vary across the laying sequence [38,39] and associations between helper numbers and egg composition have also been found to vary across the laying sequence [40], our analysis of egg volume also controls for effects of egg position within the laying sequence and allows for interactive effects of helper numbers and egg position. We then investigate whether within-mother variation in female and male helper numbers predict variation in the mother's nestling feeding rate (again utilizing a large longitudinal data set; 124 broods being fed by 50 mothers in 34 social groups; 1 to 7 broods [median = 2] per mother). Our analyses also allow for effects of variation in abiotic conditions (rainfall and temperature) on mean levels of maternal investment [41,42].

## Results

### The patterns and implications of maternal variation in egg volume

Sparrow-weavers show appreciable variation in egg volume both within and among mothers (Fig 1A). The average egg volume per mother was 3.665 cm³ (range = 2.850 cm³ to 4.462 cm³); with a maternal repeatability for egg volume of 69.3% (i.e., the amount of total variation in egg volume explained by a mother ID random effect; $\chi^2_1 = 129.89$, $p < 0.001$). Egg volume appears to provide a valid proxy for pre-natal maternal investment per offspring, as higher volume eggs were heavier at laying (effect of egg volume on egg mass ± standard error [SE] = 0.951 ± 0.018 g/cm³; $\chi^2_1 = 625.98$, $p < 0.001$; Fig 1B and S1 Table) and yielded heavier nestlings at hatching (effect of egg volume on hatchling mass ± SE = 0.679 ± 0.124 g/cm³; $\chi^2_1 = 27.23$, $p < 0.001$; Fig 1C and S2 Table). The relationship between egg volume and hatchling mass also holds within mothers, illustrating that maternal plasticity in egg volume is also a key source of variation in nestling mass at hatching (effect of within-mother variation in egg volume on hatchling mass ± SE = 0.846 ± 0.228 g/cm³; $N = 193$ eggs from 54 mothers; $\chi^2_1 = 13.22$, $p < 0.001$; S3 Table). Laying larger eggs could therefore have fitness consequences for mothers and the resulting offspring (see Discussion).

### Maternal plasticity in pre-natal investment per offspring: Individual mothers lay larger eggs when they have more female helpers

Modeling the causes of variation in egg volume revealed that mothers with more female helpers at laying laid significantly larger eggs (female helper number effect ± SE = 0.018 ± 0.009

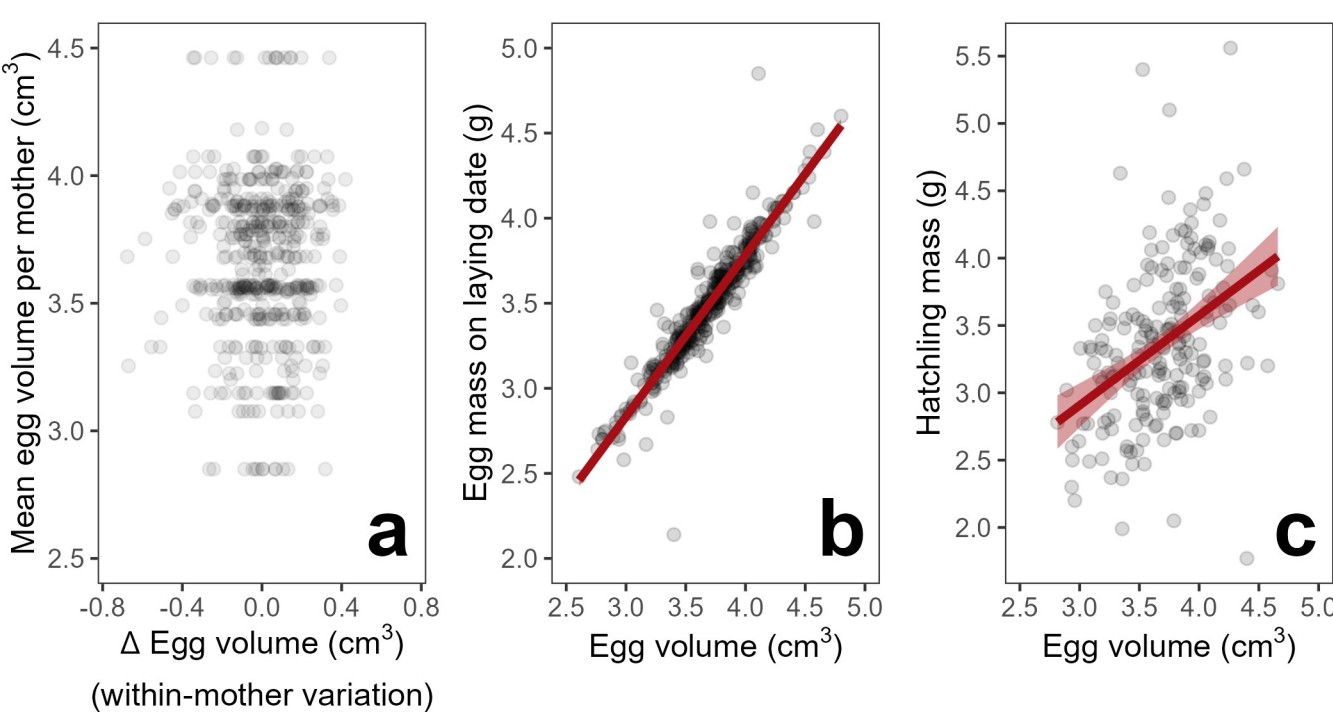

**Fig 1. Patterns and implications of maternal variation in egg volume.** (**a**) Egg volume showed high variation both within (x-axis) and among mothers (y-axis). Δ egg volume represents the difference in egg volume between the focal egg and that mother's own mean egg volume (i.e., within-mother variation, hence the negative and positive values). (**b**) Variation in egg volume positively predicted egg mass (g) on the day of laying (effect size ± standard error [SE] = 0.951 ± 0.018 g/cm³; $N = 391$ eggs with volume and laying mass data; $\chi^2_1 = 625.98$, $p < 0.001$; S1 Table) and (**c**) nestling mass (g) on the day of hatching (effect size ± SE = 0.679 ± 0.124 g/cm³; $N = 193$ eggs with volume and hatchling mass data; $\chi^2_1 = 27.23$, $p < 0.001$; S2 Table). Mean model predictions ± SE are plotted in red. The data underlying this figure can be found at https://doi.org/10.5281/zenodo.8385995.

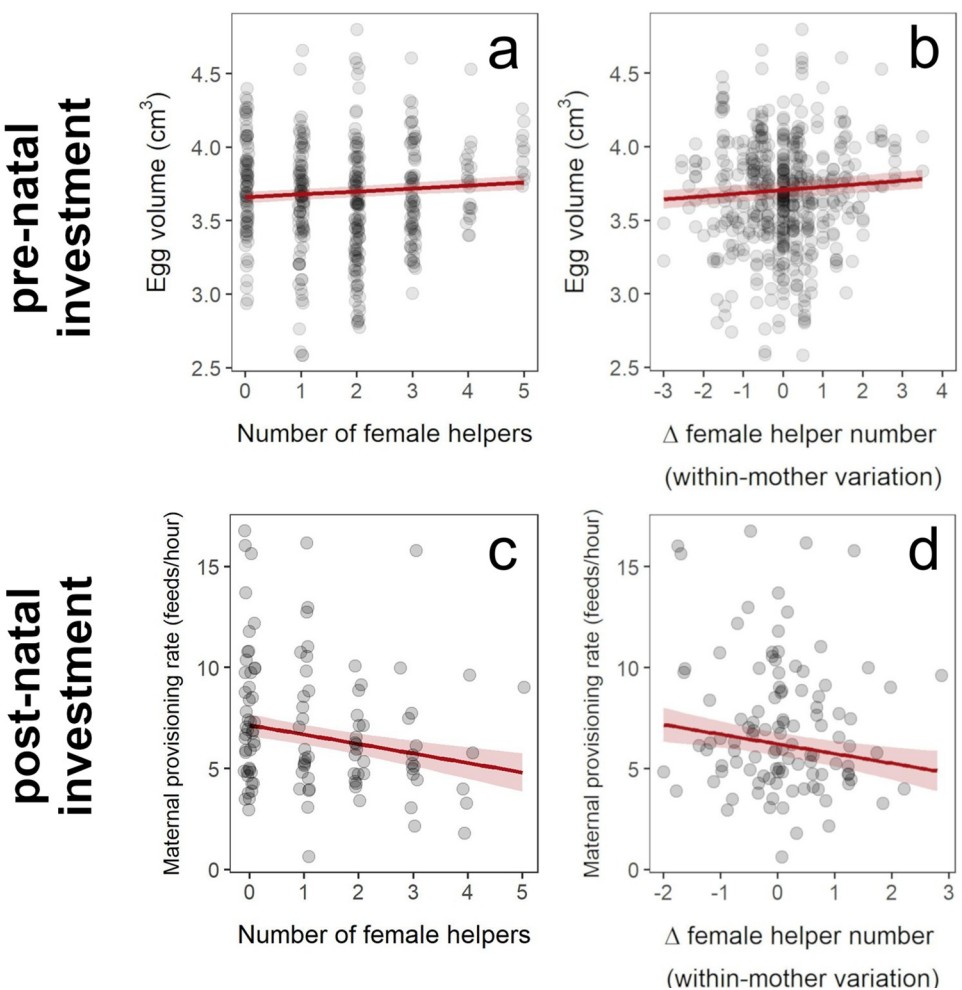

**Fig 2. Maternal plasticity in pre-natal (egg volume) and post-natal (nestling provisioning rate) investment according to female helper numbers.** (a) Female helper number positively predicts egg volume at the population level (Table 1; prior to partitioning variation in helper number). (b) Within-mother variation in female helper number ("Δ female helper number") also positively predicts egg volume, providing evidence of maternal plasticity (see Results, Tables 2 and S5). (c) Female helper number negatively predicts maternal nestling provisioning rate at the population level (Table 3). (d) Within-mother variation in female helper number ("Δ female helper number") also negatively predicts maternal nestling provisioning rate, providing evidence of maternal plasticity (see Results, Tables 4 and S11). Gray dots illustrate raw data points, and red lines present model predictions (± SE). The data underlying this figure can be found at https://doi.org/10.5281/zenodo.8385995.

cm$^3$/female helper; $\chi^2_1 = 4.31$, $p = 0.038$; Fig 2A and Table 1). To identify whether this population-level relationship arose in part from a maternal plastic response to female helper numbers, we repeated the analysis after partitioning variation in female (and male) helper numbers into their within-mother ("Δ female helper number") and among-mother ("μ female helper number") components. This analysis revealed evidence of maternal plasticity in egg volume according to female helper numbers (i.e., a maternal reaction norm to variation in female helper number): within-mother variation in female helper number significantly positively predicted egg volume (Δ female helper number effect ± SE = 0.019 ± 0.009 cm$^3$/female helper; $\chi^2_1 = 4.36$, $p = 0.037$; Fig 2B and Table 2). This partitioning approach is expected to yield unbiased estimates of within-individual effects [43], but we also confirmed that a more data-demanding bivariate approach yielded a similar effect size estimate for the within-mother reaction norm

**Table 1. Summary of results of a linear mixed model explaining variation in egg volume (cm$^3$) and including every main effect of interest ($N$ = 490 eggs laid in 271 clutches by 62 mothers in 37 social groups; 1–21 eggs [median = 7] measured per mother).** Model estimates, standard errors (SE), and their 95% confidence intervals (CI (95%)) are provided along with results from likelihood-ratio tests ($\chi^2_{df = 1}$ and associated $p$-values) assessing the statistical significance of each predictor within the full model (i.e., a model containing all of the terms in the table below). Random effect standard deviation: "season" = 0 cm$^3$, "group ID" = 0 cm$^3$, "clutch ID" = 0.070 cm$^3$, "mother ID" = 0.275 cm$^3$. "Heat waves" (days above 35°C), "Clutch size" and "Egg position" were mean centered and scaled by one standard deviation prior model fit to improve model convergence. Estimates for "Rainfall" and "Rainfall$^2$" given for 100 mm of rainfall (e.g., change in egg volume [cm$^3$] per 100 mm of rainfall). "df" = degrees of freedom for likelihood-ratio tests. This analysis yielded the same conclusions when taking an AIC-based model selection approach (S4 Table).

| Predictors | Estimates | SE | CI (95%) | $\chi^2_1$ | $p$-value |
|---|---|---|---|---|---|
| Intercept | 3.630 | 0.039 | 3.553, 3.707 | - | - |
| Rainfall | 0.310 | 0.116 | 0.082, 0.538 | - | - |
| Rainfall$^2$ | −0.005 | 0.001 | −0.007, −0.002 | 15.46 | <0.001 |
| Heat waves | −0.042 | 0.010 | −0.061, −0.023 | 17.89 | <0.001 |
| Number of female helpers | 0.018 | 0.009 | 0.001, 0.035 | 4.31 | 0.038 |
| Number of male helpers | 0.008 | 0.010 | −0.011, 0.026 | 0.61 | 0.435 |
| Clutch size | 0.001 | 0.011 | −0.021, 0.023 | 0.01 | 0.918 |
| Egg position | −0.043 | 0.009 | −0.061, −0.026 | 23.20 | <0.001 |

to female helper number (estimate [95% credible interval] = 0.015 [−0.004, 0.033] cm$^3$/female helper; see S1 File, Section H).

Male helper number did not significantly predict variation in egg volume, either in our initial analysis at the population level (male helper number effect ± SE = 0.008 ± 0.010 cm$^3$/male helper; $\chi^2_1$ = 0.61, $p$ = 0.435; Table 1) or following the partitioning of the helper number terms into their within- and among-mother components (Δ male helper number effect ± SE = 0.008 ± 0.010 cm$^3$/male helper; $\chi^2_1$ = 0.57, $p$ = 0.450; Table 2). The point estimate for the effect size of the maternal plastic response in egg volume to female helper number was also approximately twice that for male helper number (Table 2), though these effect sizes did not significantly differ ($\chi^2_1$ = 0.52, $p$ = 0.471). Egg volume was also predicted by the position of the egg in the laying order (the first laid egg was larger; Table 1) and by environmental temperature and rainfall (Table 1; the effects of these abiotic predictors are discussed in detail in S1 File, Section A). We found no evidence in this analysis that the magnitude of the helper

**Table 2. Summary of results of a linear mixed model explaining variation in egg volume (cm$^3$), including every main effect of interest when population-level variation in female and male helper number were partitioned into their within-mother (Δ) and among-mother (μ) components.** Note that partitioning within-mother (Δ) and among-mother (μ) components might produce bias in the among-mother component [70]; however, this method produces robust estimation of the within-mother component, which the evidence of plasticity is based upon. Sample size and structure were identical to those for Table 1. Model estimates, standard errors (SE) and their 95% confidence intervals (CI (95%)) are provided along with results from likelihood-ratio tests ($\chi^2_{df = 1}$ and associated $p$-values) assessing the statistical significance of each predictor within the full model (i.e., a model containing all of the terms in the table below). Random effect standard deviation: "season" = 0 cm$^3$, "group ID" = 0 cm$^3$, "clutch ID" = 0.070 cm$^3$, "mother ID" = 0.275 cm$^3$. "Heat waves" (days above 35°C), "Clutch size," and "Egg position" were mean centered and scaled by one standard deviation prior model fit to improve model convergence. Estimates for "Rainfall" and "Rainfall$^2$" given for 100 mm of rainfall (e.g., change in egg volume [cm$^3$] per 100 mm of rainfall). "df" = degrees of freedom for likelihood-ratio tests. This analysis yielded the same conclusions when taking an AIC-based model selection approach (S5 Table).

| Predictors | Estimates | SE | CI (95%) | $\chi^2_1$ | $p$-value |
|---|---|---|---|---|---|
| Intercept | 3.641 | 0.051 | 3.540, 3.741 | - | - |
| Rainfall | 0.313 | 0.116 | 0.084, 0.542 | - | - |
| Rainfall$^2$ | −0.005 | 0.001 | −0.007, −0.002 | 15.58 | <0.001 |
| Heat waves | −0.042 | 0.010 | −0.061, −0.023 | 17.77 | <0.001 |
| Δ Number of female helpers | 0.019 | 0.009 | 0.001, 0.037 | 4.36 | 0.037 |
| μ Number of female helpers | 0.009 | 0.028 | −0.045, 0.063 | 0.10 | 0.755 |
| Δ Number of male helpers | 0.008 | 0.010 | −0.012, 0.028 | 0.57 | 0.450 |
| μ Number of male helpers | 0.008 | 0.029 | −0.048, 0.065 | 0.09 | 0.769 |
| Egg position | −0.043 | 0.009 | −0.061, −0.026 | 23.18 | <0.001 |
| Clutch size | 0.001 | 0.011 | −0.021, 0.023 | 0.01 | 0.909 |

number effects on egg volume depended on clutch size or egg position for either female or male helper numbers (for all interactions $\chi^2_1 < 0.26$, $p > 0.593$). An extended analysis, including additional data from eggs for which we did not know the egg's position within the clutch's laying order (and hence could not account for the significant effects of egg position or tease apart clutch size and egg position effects), highlights that the magnitude of the maternal reaction norm of egg volume to female helper number might depend upon clutch size and/or egg position, but whether this reflects biological reality is unclear given weaknesses in this expanded data set (see S1 File, Section J).

Our analyses also allowed for an effect of clutch size on egg volume, but no such association was detected (effect of clutch size on egg volume ± SE = 0.001 ± 0.011 cm$^3$/per egg in the clutch; $\chi^2_1 = 0.01$, $p = 0.918$; Table 1). Separate analyses also revealed no evidence that mothers adjust their clutch size according to helper numbers. Analysis at the population level revealed that clutch size was not significantly predicted by either the number of female helpers at laying (effect size ± SE = −0.046 ± 0.041 eggs/female helper, $\chi^2_1 = 1.29$, $p = 0.256$; S6 Table; $N = 344$ clutches laid by 66 mothers in 37 social groups; 1 to 15 clutches [median 4] per mother) or the number of male helpers at laying (effect size ± SE = −0.043 ± 0.047 eggs/male helper, $\chi^2_1 = 0.85$, $p = 0.357$; S6 Table). We found similar results after partitioning variation in helper numbers; clutch size was not significantly predicted by within-mother variation in either female helper number (Δ female helper number effect ± SE = −0.047 ± 0.051 eggs/female helper; $\chi^2_1 = 0.85$, $p = 0.358$; S7 Table) or male helper number (Δ male helper number effect ± SE = −0.049 ± 0.060 eggs/male helper; $\chi^2_1 = 0.66$, $p = 0.416$; S7 Table). The order of the clutch within the breeding season (e.g., a mother's first, second or third clutch) did not explain variation in clutch size either (effect of clutch order on clutch size ± SE = 0.032 ± 0.046 eggs/clutch; $\chi^2_1 = 0.50$, $p = 0.478$; S6–S9 Tables).

## Maternal plasticity in post-natal investment: Individual mothers provision nestlings at lower rates when they have more female helpers

Mothers that had more female helpers during the nestling period provisioned their nestlings at significant lower rates (female helper number effect ± SE = −0.457 ± 0.195 feeds/hour/female helper; $\chi^2_1 = 5.39$, $p = 0.020$; Fig 2C and Table 3). Partitioning the female helper number predictor into its within- and among-mother components revealed evidence of maternal plasticity in nestling provisioning rate according to female helper numbers (i.e., a maternal reaction norm to variation in female helper number): within-mother variation in female helper number

**Table 3. Summary of results of a linear mixed model explaining variation in maternal provisioning rate (feeds/hour) and including every main effect of interest** (*N* = 124 broods being fed by 50 mothers in 34 social groups; 1–7 broods [median = 2] per mother). Model estimates, standard errors (SE), and their 95% confidence intervals (CI (95%)) are provided along with results from likelihood-ratio tests ($\chi^2_{df = 1}$ and associated *p*-values) assessing the statistical significance of each predictor within the full model (i.e., a model containing all of the terms in the table below). Random effect standard deviation: "season" = 0.97 feeds/hour, "group ID" = 0 feeds/hour, "mother ID" = 0 feeds/hour. "Heat waves" (days above 35˚C) and "Brood size" were mean centered and scaled by one standard deviation prior model fit to improve model convergence. Estimates for "Rainfall" and "Rainfall$^2$" given for 100 mm of rainfall (e.g., change in maternal provisioning rate [feeds/hour] per 100 mm of rainfall). "df" = degrees of freedom for likelihood-ratio tests. This analysis yielded the same conclusions when taking an AIC-based model selection approach (S10 Table).

| Predictors | Estimates | SE | CI (95%) | $\chi^2_{df = 1}$ | *p*-value |
|---|---|---|---|---|---|
| Intercept | 7.460 | 0.537 | 6.325, 8.559 | - | - |
| Rainfall | −7.974 | 4.055 | −16.117, 0.218 | - | - |
| Rainfall$^2$ | 0.201 | 0.072 | 0.057, 0.345 | 7.34 | 0.007 |
| Heat waves | 0.699 | 0.302 | 0.092, 1.296 | 5.03 | 0.025 |
| Number of female helpers | −0.457 | 0.195 | −0.841, −0.073 | 5.39 | 0.020 |
| Number of male helpers | −0.072 | 0.228 | −0.530, 0.381 | 0.10 | 0.754 |
| Brood size | 1.438 | 0.244 | 0.952, 1.920 | 30.02 | <0.001 |

**Table 4. Summary of results of a linear mixed model explaining variation in maternal provisioning rate (feeds/hour), including every main effect of interest when population-level variation in female and male helper number were partitioned into their within-mother (Δ) and among-mother (μ) components.** Note that partitioning within-mother (Δ) and among-mother (μ) components might produce bias in the among mother component [70]; however, this method produces robust estimation of the within-mother component, which the evidence of plasticity is based upon. Sample size and structure were identical to those for Table 3. Model estimates, standard errors (SE), and their 95% confidence intervals (CI (95%)) are provided along with results from likelihood-ratio tests ($\chi^2_{df = 1}$ and associated $p$-values) assessing the statistical significance of each predictor within the full model (i.e., a model containing all of the terms in the table below). Random effect standard deviation: "season" = 0.92 feeds/hour, "group ID" = 0 feeds/hour, "mother ID" = 0 feeds/hour. "Heat waves" (days above 35˚C) and "Brood size" were mean centered and scaled by one standard deviation prior model fit to improve model convergence. Estimates for "Rainfall" and "Rainfall²" given for 100 mm of rainfall (e.g., change in maternal provisioning rate [feeds/hour] per 100 mm of rainfall). "df" = degrees of freedom for likelihood-ratio tests. This analysis yielded the same conclusions when taking an AIC-based model selection approach (S11 Table).

| Predictors | Estimates | SE | CI (95%) | $\chi^2_{df = 1}$ | $p$-value |
|---|---|---|---|---|---|
| Intercept | 7.445 | 0.641 | 6.130, 8.750 | - | - |
| Rainfall | −8.042 | 4.053 | −16.164, 0.127 | - | - |
| Rainfall² | 0.203 | 0.072 | 0.059, 0.347 | 7.50 | 0.006 |
| Heat waves | 0.713 | 0.302 | 0.105, 1.310 | 5.21 | 0.022 |
| Δ Number of female helpers | −0.559 | 0.269 | −1.090, −0.027 | 4.24 | 0.040 |
| μ Number of female helpers | −0.325 | 0.304 | −0.928, 0.287 | 1.14 | 0.285 |
| Δ Number of male helpers | 0.006 | 0.307 | −0.601, 0.612 | <0.01 | 0.986 |
| μ Number of male helpers | −0.188 | 0.408 | −1.042, 0.620 | 0.21 | 0.646 |
| Brood size | 1.446 | 0.244 | 0.959, 1.928 | 30.26 | <0.001 |

significantly negatively predicted a mother's provisioning rate (Δ female helper number effect ± SE = −0.559 ± 0.269 feeds/hour/female helper; $\chi^2_1$ = 4.24, $p$ = 0.040; Fig 2D and Table 4). Again, we confirmed that a more data-demanding bivariate approach yielded a similar effect size estimate for the within-mother reaction norm to female helper number (estimate [95% credible interval] = −0.451 [−1.077, 0.151] feeds/hour/female helper; see S1 File, Section H).

Male helper number did not significantly predict variation in maternal provisioning rate, either in our initial analysis at the population level (male helper number effect ± SE = −0.072 ± 0.228 feeds/hour/male helper; $\chi^2_1$ = 0.10, $p$ = 0.754; Table 3) or following the partitioning of the helper number terms into their within- and among-mother components (Δ male helper number effect ±SE = 0.006 ± 0.307 feeds/hour/male helper; $\chi^2_1$ = < 0.01, $p$ = 0.986; Table 4). The point estimate for the effect size of the maternal plastic response in provisioning rate to female helper number was also larger than that for male helper number (Table 4), though these effect sizes did not significantly differ ($\chi^2_1$ = 1.57, $p$ = 0.210). Maternal nestling provisioning rates were also significantly positively related to brood size (brood size effect on maternal provisioning rate = 1.438 ± 0.244 feed/hour/nestling; $\chi^2_1$ = 30.02, $p$ < 0.001; Table 3) and were predicted by environmental temperature and rainfall (Table 3; discussed in detail in S1 File, Section A). We found no evidence that the magnitude of the helper number effects on maternal provisioning rate depended on brood size for either female or male helper numbers (for all interactions $\chi^2_1$ < 2.10, $p$ > 0.148).

## Discussion

To test the predictions of the "load-lightening" and "differential allocation" hypotheses for the evolution of pre-natal investment strategies in cooperative breeders, we investigated the patterns of maternal plasticity in both pre- and post-natal investment per offspring in white-browed sparrow-weaver societies. Using a within-mother reaction norm approach, our analyses revealed the first formal evidence of maternal plasticity in egg investment according to the availability of help in a cooperatively breeding bird (see Introduction and [29]). When sparrow-weaver mothers had more female helpers, they laid modestly but significantly larger eggs. As larger eggs also yield heavier hatchlings, this plastic response likely does reflect a change in

maternal investment and has the potential to impact offspring fitness. This positive maternal plastic response runs counter to the leading "load-lightening hypothesis" (which predicts that helped mothers should lay smaller eggs; [6,7]) and counter to general expectation given empirical work to date [13]. The "differential allocation hypothesis" does predict that helped mothers should lay larger eggs (as we observe) but is thought to predict that helped mothers should also feed their nestlings at higher rates (i.e., mothers should increase both pre- and post-natal investment per offspring when helped; see "silver spoon" scenario in [6]). By contrast, our findings reveal a novel maternal strategy in which mothers with more (female) helpers appear to increase pre-natal investment per offspring (lay larger eggs) but decrease post-natal investment per offspring (feed their nestlings at lower rates). We term this strategy "maternal front-loading," as mothers effectively front-load their investment to the pre-natal stage when helped. We consider adaptive explanations for this strategy below, along with its implications for identifying the benefits of helping in cooperative societies. That sparrow-weaver mothers increase pre-natal investment per offspring when helped highlights the potential for post-natal helping to have beneficial effects on the pre-natal development of young.

While relationships between helper number and egg size have previously been reported in cooperatively breeding birds ([13]; and see Introduction), our findings evidence that such a pattern can arise from within-mother plasticity. This is important as recent work has highlighted that population-level relationships between helper number and egg size (i.e., those reported to date: e.g., [7,8,11,12,21]) can arise from among-mother variation in egg size rather than within-mother plasticity [10]. While the effect size for the apparent maternal plastic response to female helper numbers observed here is modest, it is of broadly comparable magnitude to the effect size previously reported for the population-level relationship between helping and egg volume in the study that initially stimulated research this field ([7]; S1 File, Section B), but differs in that it captures the within-mother plastic response, and is in the opposite direction. That sparrow-weaver mothers appear to significantly adjust egg size according to female helper number and not male helper number implicates the availability of post-natal helping per se as the likely driver of this plastic maternal response, rather than correlated variation in group size (as female helpers feed nestlings at twice the rate of male helpers, and only female helper number has a demonstrably causal positive effect on the overall rate of nestling provisioning; [35]). Indeed, as female and male helper numbers at laying strongly predict helper numbers during the post-natal care period (S1 Fig), sparrow-weaver mothers should have sufficient information at laying to adjust their egg volume to the future availability of post-natal help, were it adaptive to do so.

Our use of the within-mother reaction norm approach ensures that the apparent effects of female helper numbers on both egg volume and maternal provisioning rate cannot be attributed instead to correlated among-mother variation in either maternal or territory quality. Nor can they be readily attributed instead to correlated temporal variation in abiotic factors such as rainfall or temperature, as our analyses simultaneously allow for independent effects of rainfall and temperature (which our sliding window approach allows to take many forms; see Materials and methods). Furthermore, our findings cannot be attributed to a confounding correlation between helper numbers and maternal age as these 2 variables are not correlated in our study population [35,44]. The positive within-mother relationship between female helper number and egg volume could conceivably emerge as a by-product of a helper effect on the mother's optimal clutch size or number of clutches per year, with which egg volume could trade off [10]. This mechanism cannot readily account for our findings, however, as sparrow-weaver mothers vary neither clutch size nor clutch number according to helper numbers (see Results and S1 File, Section C). Additional analyses also suggest that maternal plasticity in egg

volume cannot be readily attributed to carry over effects on maternal condition of helper actions in previous breeding attempts (see S1 File, Sections D-G).

While the "differential allocation hypothesis" does predict the pattern of plasticity in pre-natal investment observed here (mothers increase egg volume when helped), the observed pattern of plasticity in post-natal investment (mothers decrease nesting provisioning rate when helped) runs counter to that recently predicted under differential allocation [6]. In general terms, the differential allocation hypothesis proposes that mothers should increase maternal investment under circumstances that increase their return on investment in the current breeding attempt, such as having a more attractive mate or more helpers [4,24]. Accordingly, models that apply this rationale specifically to pre-natal investment in cooperative breeders (by having the mother's return on pre-natal investment per offspring increase when she has help with post-natal care; [6]) predict that mothers should increase both their pre- and post-natal investment per offspring when helped. These predictions are consistent with the patterns observed in one of the 2 other species to date in which mothers are thought to consistently lay larger eggs when they have more help: Iberian magpie mothers with helpers appear to lay larger eggs and provision their nestlings at higher rates than those without helpers ([21,25]; but whether either reflects maternal plasticity is unknown). It is notable then that sparrow-weaver mothers instead increase pre-natal investment per offspring while decreasing post-natal investment per offspring when helped (i.e., engage in "maternal front-loading," a pattern that may also occur in placid greenbuls [21], but whether it reflects plasticity in that species is unknown). Despite this discord, it would seem premature to rule out a role for differential allocation in the maternal strategy observed here, as the relevant theoretical work to date [6] might not capture all relevant aspects of the biology at play. For example, as per the differential allocation rationale, sparrow-weaver mothers might increase egg size with female helper number because the additive post-natal care that their female helpers will provide (i.e., the net-positive effect of female helpers on the overall rate at which broods are provisioned; [35]) increases the mother's expected return on investment per egg (e.g., producing larger hatchlings may yield a greater payoff when they stand to be fed at higher rates [6,8]). Such "additive" post-natal care by female helpers [35] appears to be accompanied here by mothers decreasing their own post-natal contributions when helped, a maternal strategy of post-natal "partial compensation" commonly observed in cooperative breeders [5,15]. As such, modifications to existing models to incorporate, or more fully explore, the selective pressures that favor such maternal post-natal compensation (e.g., strongly diminishing returns of post-natal care [5,45]) could conceivably leave the differential allocation rationale predicting the maternal strategy observed here (i.e., differential allocation at the egg stage accompanied by maternal post-natal compensation). The integration of stronger maternal trade-offs between pre- and post-natal investment and/or higher costs to mothers of post-natal investment might also resolve the apparent discord.

While the differential allocation rationale could conceivably explain the maternal investment strategy observed here (see above), our findings do highlight a simpler explanation for sparrow-weaver mothers laying larger eggs when helped. The differential allocation hypothesis envisages that helpers increase the maternal benefit of pre-natal investment per offspring (e.g., via the provision of additive post-natal care; [6]). However, helpers may instead reduce the maternal cost of pre-natal investment per offspring by reducing maternal post-natal workloads. Maternal front-loading may therefore reflect an anticipatory strategy in which the expected lightening of maternal post-natal workloads allows helped mothers to focus their investment on the pre-natal phase, to which helpers cannot contribute directly. Such a maternal strategy may therefore be of particular benefit when pre-natal investment has differentially large effects on offspring fitness. Under this scenario, the maternal increase in egg investment

when helped is a consequence of the helper effect on the mother's post-natal workload, whereas under the differential allocation hypothesis, the increase is typically considered a product of the additive (i.e., net positive) effect of helpers on the overall provision of post-natal care [4,8,13,21]. Species in which helpers lighten maternal post-natal workloads but do not have additive effects on post-natal care (because the maternal reduction in post-natal work rate completely compensates for helper contributions [5]) would therefore provide a fruitful testing ground for these alternative, though not mutually exclusive, hypotheses. As helpers frequently lighten maternal post-natal workloads in cooperative breeders [5,7,45,46], the maternal front-loading strategy observed here could ultimately prove more commonplace once more studies formally characterize maternal plasticity in egg investment [10]. For example, recent evidence suggesting that superb fairywren mothers with helpers may lay larger eggs than those without when conditions are warm [8] could reflect maternal front-loading in warm conditions, if the reported population-level relationship between egg size and the availability of help arose via maternal plasticity, and if post-natal load-lightening also occurred under such warm conditions (which it might [47]). While red-winged fairywren mothers do not increase egg size when helped, they do show a plastic increase in clutch size when helped [10], another form of maternal front-loading that could arise via the same mechanism: helpers reducing the cost of egg investment by lightening the mother's post-natal workload [48].

This hypothesis that post-natal load-lightening by helpers reduces the costs to the mother of pre-natal investment in the egg is predicated upon the expectation of a resource allocation trade-off between maternal pre- and post-natal investment, which theory expects and there is experimental empirical data to support [49–51]. While this could certainly be the case in sparrow-weavers, an attempt to characterize this trade-off via a bivariate analysis of natural variation in pre- and post-natal maternal investment using our data did not yield evidence of such a trade-off (S1 File, Section I). This is perhaps not surprising though, as experimental manipulations (e.g., of maternal pre-natal investment in this context) are often required to expose trade-offs that may otherwise be shrouded by confounding natural variation in overall resource availability between contexts [52,53]. For example, in our study, natural variation in rainfall-related resource availability among breeding attempts could leave natural variation in pre- and post-natal maternal investment positively correlated within mothers even if such a trade-off exists. It is nevertheless conceivable that such a trade-off is not sufficiently strong to explain our findings from an adaptive perspective. Indeed, while the field is focused on adaptive explanations for plasticity, we should acknowledge too the wider possibility that the net fitness consequences of the observed maternal plastic response are not sufficiently strong across contexts for the response to have been optimized by selection, such that adaptive explanations may not apply.

The 2 potential adaptive explanations that we consider above for the observed maternal plasticity in egg volume both assume that increasing egg volume increases offspring fitness (in at least some social contexts; see below). Beneficial effects of maternal investment at the egg stage are to be expected, given the wealth of evidence to this effect from other species [1–3]. Our findings also illustrate one plausible path for such fitness effects in sparrow-weavers; here, we show that larger eggs yield heavier hatchlings (Fig 1), and we have previously shown that heavier hatchlings are more likely to survive to fledging (S3B Fig in [35]). That said, a previous analysis using this species did not detect a direct relationship between the average egg volume of a clutch and offspring survival to fledging from that clutch (S3A Fig in [35]), leaving it unclear to what extent egg volume impacts offspring fitness through this specific path or others (e.g., via effects on the downstream survival or performance of fledged young). In reality, the relationship between egg volume and offspring fitness could well be complex, as under adaptive explanations for maternal plasticity in egg volume the fitness implications of variation in

egg volume are expected to be context dependent. For example, the rationale of the differential allocation hypothesis (that post-natal care from helpers increases the payoffs to mothers from pre-natal investment in the egg) assumes not a uniformly beneficial effect of egg investment, but a more complex interactive effect in which the benefits of egg investment depend upon the social environment. Our ability to detect potentially subtle or interactive causes of variation in offspring survival may be hampered in this species by survival to fledging being driven by strongly interactive effects of rainfall and female helper numbers [35]. Nevertheless, a priority for future research will be investigating the effects of the substantial variation in egg volume that we observe in this species on the early-life and downstream performance of offspring.

Where mothers do increase their pre-natal investment per offspring when helped (as observed here), post-natal helping may have hitherto unexplored beneficial effects on the pre-natal development of offspring. The potential for such cryptic "pre-natal helper effects" has important implications for attempts to identify and quantify the benefits of helping in cooperative societies. First, while it has been suggested that studies of helper effects on offspring should control for variation in egg size in order to ensure that maternal reductions in egg size by helped mothers do not "conceal" helper effects on offspring [7], our findings highlight the need for caution with this approach. If, as here, mothers lay larger eggs when helped, controlling for variation in egg size could lead to the underestimation of the total helper effect on offspring, by factoring out helper effects that arise indirectly via maternal investment in the egg. While we cannot formally assess the relative importance for offspring fitness of this potentially "indirect" helper effect on maternal pre-natal investment and the direct causal positive effect of helpers on post-natal provisioning (demonstrated in [35]), the direct post-natal helper effect may still dominate, as the within-mother effect of female helper number on egg volume is modest compared to the within-mother effect of female helper number on the overall rates of post-natal provisioning (Fig 1 in [35]). That said, the effect size for the change in egg volume could underestimate the fitness consequences for offspring of the maternal pre-natal response to helpers, as any accompanying helper-induced changes in egg composition [7,40] could yield fitness effects that differ in magnitude from the observed change in egg volume. Second, while helper-induced reductions in maternal post-natal workloads are typically thought to benefit mothers (e.g., by improving maternal survival; [5,7]), our findings highlight that associated changes in egg investment could pass these benefits, in part or whole, to the offspring being reared. Indeed, as helpers commonly lighten maternal post-natal workloads [5,45–47], a maternal front-loading response of the type observed here could conceivably have contributed to the positive relationships already described in numerous species between helper numbers and offspring survival or performance.

Our findings provide formal evidence of maternal plasticity in pre-natal investment per offspring according to the availability of help in a natural population [29]. They reveal a plastic maternal pre-natal response that runs counter to the predictions of the leading load-lightening hypothesis and to general expectation given the limited empirical work to date [13]. The patterns of maternal plasticity in post-natal investment that we also document suggest that the overall maternal strategy does not match the existing predictions of the differential allocation hypothesis either [6] and instead highlight an alternative explanation for mothers increasing their egg size when helped: By lightening maternal post-natal workloads, helpers may allow mothers to focus their investment on the pre-natal stage, to which helpers cannot contribute directly. That mothers increased pre-natal investment per offspring when helped highlights the potential for post-natal helping to promote the pre-natal development of offspring. The potential for such cryptic maternally mediated helper effects on pre-natal development may also markedly complicate attempts to identify and quantify the fitness consequences of helping.

## Materials and methods

### Ethics statement

Animal ethics approval for this study was granted by ethics committees at the University of Pretoria and the University of Exeter. Bird ringing was carried out under SAFRING license 1444.

### General field methods

White-browed sparrow-weavers live in semi-arid regions of East and Southern Africa. Our study population is located in Tswalu Kalahari Reserve in the Northern Cape Province of South Africa (27˚ 16' S, 22˚ 25' E). Fieldwork was carried out from September to April between 2007 and 2016 inclusive. Approximately 40 social groups were monitored, each defending a small exclusive territory within an overall study site of approximately 1.5 km². Sparrow-weaver groups were easily monitored and distinguished in the field as all group members foraged together, engaged in communal weaving and territory defense, and roosted within in a single tree or cluster of trees close to the center of their territory. All birds in the study population were fitted with a single metal ring and 3 color rings for identification from the time they were first detected in the population (under SAFRING license 1444). The sex of each bird could be determined after 6 months of age using the sex difference in bill color [54].

Each social group contains a single behaviorally dominant female. The dominant female is easily identified in the field because she displays a distinct set of behaviors: being behaviorally dominant to other females, being the only female observed to incubate the eggs or enter the nest during the incubation phase, and closely associating with and frequently duetting with the dominant male [55]. Genetic analyses have confirmed that the dominant female is always the mother of any eggs or chicks produced on their group's territory; subordinate females never breed [33]. For brevity, we therefore refer throughout the paper to the dominant female as the "mother."

Each group's territory was regularly monitored (every 1 or 2 days while nests were present) to detect new clutches. Once a new clutch was found, daily checks were made for new eggs until the clutch was complete to determine the position of each egg within the laying order. On discovery, each new egg was individually marked with a non-toxic marker, measured (egg length and maximum width) with a plastic caliper to the nearest 0.1 mm, and weighed using a portable scale to the nearest 0.01 g. Clutches were then checked 8 days after the first egg was laid (to confirm the progression of incubation), before daily checks were resumed 15 days after the first egg was laid, until the fate of every egg had been determined (hatch or failure). Modest hatching asynchrony is common in white-browed sparrow-weaver broods, and so it was often possible to link individual eggs to individual hatchlings directly (e.g., when one egg in the modal clutch size of 2 had not yet hatched). Hatchlings were weighed on their first day of life using a portable scale to the nearest 0.01 g.

The composition of each social group was assessed every week throughout each field season, with birds being identified on the basis of their color-ring combination. Birds were also routinely caught while roosting within their group's territory at night, and this information also contributed to group composition assessments. Group compositions were typically very stable over time, with group members residing within the same social group for many months to many years at a time (i.e., group composition not being affected by short-term fluctuations in environmental conditions). For every breeding attempt in our analyses, we used these group compositions to calculate the number of male and female helpers that the dominant female (mother) had on the day of laying (for the egg volume analyses) and on the days that

provisioning behavior was recorded (for the maternal provisioning rate analyses). All subordinate group members over the age of 6 months were considered helpers, as analyses of helper contributions suggest that subordinates <6 months old contribute little to nestling provisioning [34,56].

## Nestling provisioning behavior

Nestling provisioning behavior was recorded between September 2007 and April 2016. We collected provisioning data using video recordings of the birds visiting the nest (viewed from below the nest) between the 10th and 12th day inclusive after the first egg of a given clutch had hatched (this is the period of highest nestling post-natal demand; the nestling period lasts approximately 20 to 25 days). At least 5 days before video recording started, we (i) caught and marked the vent of each group member other than the dominant female using hair dye [35] to aid their identification on the video and (ii) deployed a tripod on the ground beneath the nest to acclimatize the birds to its presence prior to recording. On recording days, the video camera was set up and recording started soon after sunrise, at standard times relative to sunrise in order to track seasonal changes in sunrise timings. Provisioning behavior was recorded for approximately 3 hours per day per brood. Video recordings were then watched using VLC media player to determine the rate at which each group member visited the nest (here after their "provisioning rate"), identifying each visitor via their sex (based on bill coloration [54]), unique vent pattern, and color-ring combination. Prior analyses using within-nest cameras have confirmed that, during this peak provisioning period, all nest visits by all group members entail the delivery of a single food item by the visitor that is then eaten by the chicks (the only exception being nest-maintenance visits that were easily excluded from the data set on the basis of the visitor conspicuously carrying grass [57]).

We then calculated the provisioning rate of mothers (feeds/hour). In some cases, we were unable to reliably identify every visiting bird within the provisioning video, yielding some uncertainty in our estimate of maternal provisioning rate. We therefore only carried forward maternal provisioning rate estimates to our statistical analyses where the maximum possible maternal provisioning rate (i.e., if one considered the mother the feeder in all cases of uncertain feeder identity) did not exceed the observed maternal provisioning rate (calculated solely on the basis of the mother's identified visits) by more than 3 feeds/hour. Applying this filtering criteria, there was less than 10% uncertainty for more than 90% of maternal provisioning rate estimations. Where estimates of maternal provisioning rate were available for multiple mornings for a given breeding attempt, the measures were averaged to yield a single mean maternal provisioning rate for each breeding attempt for analysis (as maternal provisioning rate estimates for a given breeding attempt were correlated over successive mornings of video recording; correlation coefficient [95% CI] = 0.46 [0.31, 0.59]). This yielded a data set for analysis of mean maternal provisioning rate for 50 different dominant females (mothers) feeding 124 broods in 34 social groups.

## Environmental data

Daily rainfall data were collected from 2 rainfall gauges located to the west (27° 16' 58.9" S, 22° 23' 02.1" E) and east (27° 17' 42.1" S, 22° 27' 34.9" E) of the study site, 7.60 km apart from each other. These 2 rainfall measurements were highly correlated during the study period (Pearson's product-moment correlation: r = 0.875, 95% CI = 0.867 to 0.882, df = 3,347). We therefore calculated average daily values across both gauges and used this as a proxy for rainfall conditions at the study site.

Temperature data for a 0.25 degree latitude × 0.25 degree longitude area that encompassed the study site was extracted from the GLDAS-2.1 Noah 0.25 degree 3-hourly data set [58], accessed via the NASA's Goddard Earth Sciences Data and Information Services Center online data system (Giovanni; http://disc.sci.gsfc.nasa.gov/giovanni). From this, we calculated the daily maximum temperature and daily mean temperature (i.e., the average of all 8 measures available per 24-hour period) for all days of our study. The daily mean temperatures from this data set were highly correlated with those obtained directly within our study site using a 2700 Watchdog weather station (Spectrum Technologies) deployed for part of the study period (partial coverage of 2010 to 2015; Pearson's product-moment correlation: r = 0.973, 95% CI = 0.970 to 0.975, df = 1,771).

## Statistical analysis

**Modeling egg volume effects on egg mass and hatchling mass within and among-mothers.** First, we investigated whether egg volume and egg mass were correlated and whether egg volume predicted hatchling mass. To this end, we fitted linear mixed models to (i) explain variation in egg mass (g) and (ii) hatchling mass (g) including egg volume as a fixed effect predictor. These models also included mother ID as a random intercept. Both models were fitted a second time to partition the effect of egg volume on each of the response variables within- and among-mothers (see details below), to investigate whether variation in egg volume within mothers (i.e., plasticity) was associated with variation in egg mass and hatchling mass (i.e., providing evidence for the biologically relevant role of plasticity in egg volume).

**Modeling maternal pre-natal investment per offspring: Egg volume.** Linear mixed models with Gaussian error structure were used to investigate the predictors of egg volume (measured in cm$^3$ and calculated based on length and maximum breadth following the formula given in [59]). Four terms were included as random intercepts: breeding season (referring to each of the 9 different September to April breeding seasons studied), social group ID, clutch ID, and maternal ID. The following were included as fixed effect predictors: egg position within the clutch, clutch size, number of female helpers, number of male helpers, and the interaction between helper number (both females and males) and (i) egg position and (ii) clutch size. These interactive terms are included to specifically control for interactive effects of helpers and egg position or clutch size on egg volume [40]. To control for the potential effects of temperature and rainfall on egg volume, we also fitted the following 2 indices as fixed effect predictors: a "heat waves" index (the number of days in which the maximum daily temperature exceeded 35˚C within a time window spanning the 13 days prior to egg laying) and a rainfall index (the total rainfall that fell within a time window spanning 44 to 49 days prior to egg laying). The specific time windows used for the calculation of these indices were determined objectively by the application of a sliding window approach prior to this modeling step (see S1 File, Section A). The "heat waves" index as defined here (i.e., number of days above 35˚C) has been shown to appropriately capture hot-weather events in the Kalahari, and it impacts the reproductive biology of several Kalahari bird species [60,61]. Egg position, clutch size, and heat waves index were standardized (i.e., mean centered and divided by one standard deviation) prior to model fitting to facilitate model convergence. Similarly, to improve model convergence, orthogonal polynomials of degree 2 (i.e., quadratic effects) were calculated to model quadratic effects of rainfall index, but model estimates were back transformed for presentation purposes and provided in change of egg volume (cm$^3$) per 100 mm of rainfall. Between 2007 and 2016 inclusive, we collected egg length and width information (and therefore volume) from 906 eggs that were detected in the field with less than 4 days of uncertainty around their laying date. Given the a priori expectation that egg position could prove an important

predictor of egg volume (both as a main effect and potentially via interactions with helper numbers; [38–40]), we focused our analysis on the 490 of these for which we also knew laying order (allowing determination of the "egg position within the clutch" variable and accurate clutch size information): 490 eggs from 271 clutches laid by 62 dominant females (mothers) across 37 social groups (mean = 7.90 eggs per mother; median = 7 eggs per mother; range 1 to 21 eggs per mother; (S6 and S7 Figs). Indeed, this analysis revealed that egg position within the clutch was the most important predictor of egg volume (see Results), suggesting that it was appropriate to apply this data quality restriction. Nevertheless, having found evidence for maternal plasticity in egg volume accordingly to female helper numbers when using this data set (see Results), we confirmed that evidence of maternal plasticity in egg volume was still apparent when conducting an "extended analysis" using the full data set ($n$ = 906 eggs; see above), despite the inability to fit egg position as a predictor when using this data (as egg position was unknown for 416 of these eggs). This extended analysis (reported in full in S1 File, Section J) (i) revealed statistical support for maternal plasticity in egg volume according to female helper numbers (egg volume was predicted by a significant interaction between within-mother variation in female helper number and clutch size) and (ii) highlighted the possibility that the positive maternal reaction norm of egg volume to female helper number is also context dependent: larger in magnitude than the effect reported in our main analyses in some clutch sizes and not discriminable from zero in others (see S1 File, Section J). Whether this interaction reflects biological reality is unclear, however, as this extended analysis cannot account for the demonstrably important effect of egg position on egg volume (see Results). It is also difficult to interpret this interaction, as the uncontrolled variation in egg position within this extended analysis will also confound variation in clutch size (e.g., because third-laid eggs only occur in 3-egg clutches), leaving it unclear whether any such interaction is driven principally by variation in clutch size or egg position effects.

**Modeling maternal post-natal investment: Maternal nestling provisioning rate.** Linear mixed models with Gaussian error structure were used to investigate the predictors of maternal provisioning rate (calculated as a single mean value for each breeding attempt; see above). Three terms were included as random intercepts: breeding season (see above), social group ID, and maternal ID. The following were included as fixed effect predictors: brood size, number of female helpers, number of male helpers, and the interactions between helper number (both females and males) and brood size. These interactive terms are included to specifically test whether the effect of helpers is dependent on brood size. To control for the potential effects of temperature and rainfall on maternal provisioning rate, we also fitted the following 2 indices as fixed effect predictors: "heat waves" index (the total number of days within a time window spanning 51 to 58 days prior to egg laying in which the maximum daily temperature exceeded 35°C) and a rainfall index (the total amount of rainfall that fell within a time window spanning 61 to 78 days prior to egg laying). The specific time windows used for the calculation of these indices were determined by the application of a sliding window approach prior to this modeling step (see S1 File, Section A). Brood size and heat waves index were standardized (i.e., mean centered and divided by one standard deviation) prior to model fitting to facilitate model convergence. Similarly, to improve model convergence, orthogonal polynomials of degree 2 (i.e., quadratic effects) were calculated to model quadratic effects of rainfall index, but model estimates were back transformed for presentation purposes and provided in change of maternal provisioning rate (feeds/hour) per 100 mm of rainfall. The final data set contained 124 measures of mean maternal provisioning rate for 124 broods born to 50 dominant females (mothers) across 34 social groups. The data set contained more than one brood for 34 mothers, while 16 mothers were observed provisioning only once (S7 Fig).

Following [62], we checked that the findings of the Gaussian modeling exercise above for maternal provisioning rate were robust to directly modeling the underlying Poisson process. Following the recommendations in [62], we reran our initial maternal provisioning rate models using a Poison GLMM, with the daily count of maternal feeds as the response; clutch ID, group ID, season, maternal ID as random intercepts. We also included an observation level random effect to account for overdispersion. We included the same fixed effects that we specified for the maternal provisioning rate model outlined above. Additionally, we included (log) video recording duration (i.e., the amount of time over which daily maternal feeds were counted) as an offset in the model. The results of this Poisson model confirmed the findings presented in the main text from the Gaussian model outlined above (S31 Table). The number of female helpers negatively predicted maternal feeding rates (now modeled as a count process, with maternal number of feeds as the response and (log) observation duration as a model offset); female helper number effect $\pm$ SE = $-0.078 \pm 0.030$, $\chi^2_1 = 6.46$, $p = 0.011$; S31 Table). Also, in line with our initial findings, this model did not yield evidence for an effect of the number of male helpers on maternal feeding rates (male helper number effect $\pm$ SE = $-0.001 \pm 0.037$, $\chi^2_1 < 0.01$, $p = 0.982$; S31 Table).

**Modeling the effect of helper numbers on clutch size.**   Generalized linear mixed models with zero-truncated Poisson error structure were used to investigate the predictors of clutch size (range 1 to 3 eggs). Zero-truncated models were fitted using the R package glmmADMB (v0.8.3.3; [63]). Three terms were included as random intercepts: breeding season (referring to each of the 9 different September to April breeding seasons studied), social group ID, and maternal ID. The following were included as fixed effect predictors: clutch order within the breeding season (a continuous variable starting from 1 for the first clutch that a given mother laid within a given breeding season), number of female helpers, and number of male helpers. The number of female and male helpers was calculated for the day on which the first egg of the focal clutch was laid (or the day at the midpoint of the window of potential lay dates for the first egg, whenever there was uncertainty regarding this lay date). The analysis used a data set of 344 clutches laid by 66 dominant females (mothers) across 37 social groups, all of which were found in the field with less than 4 days of uncertainty in the lay date of the first egg (reducing the probability that any egg in the clutch disappeared before we recorded it). Out of the 344 clutches, 284 (82.56%) were found on the day that the first egg was laid. There was 1 day of uncertainty regarding the first egg lay date for 37 clutches (10.76%), 2 days for 16 clutches (4.65%), and 3 days for 7 clutches (2.03%).

**Modeling the effect of helper numbers on the number of clutches laid per year.**   Generalized linear mixed models with Poisson error structure were used to investigate the predictors of the number of clutches that mothers laid per year (calculated as the number of clutches laid by each mother during each breeding season, running from 1 September in one calendar year to 30 April in the next; see above). For this analysis, we only used data from females that were dominant for the whole breeding season (208 clutch numbers for 56 dominant females [mothers] across 38 social groups). Three terms were included as random intercepts: breeding season, social group ID, and maternal ID. The following were included as fixed effect predictors: the mean number of female and male helpers during the focal breeding season (the average for the period 1 September to 30 April) as well as the total rainfall that fell during the focal breeding season.

**General statistical procedures.**   We built models that included fixed effect variables and interactions predicted to have an effect on the focal response term (see above for details; these were always chosen a priori based on biological hypothesis) and evaluated the statistical importance of predictors in these models via likelihood-ratio tests (LRTs). If not statistically significant, interactive terms were removed from initial models to ease the interpretation of the

effects of non-interactive terms. We provide tables of results for these full models including effect sizes (i.e., model coefficients, estimate), effect size standard errors ("SE"), model coefficient 95% confidence intervals ("95% CI"), and LRT results ($\chi^2$ value, degrees of freedom of the test, and $p$-value (Tables 1–4). We complemented this statistical approach with another analysis based on Akaike's information criterion (AIC) model selection. Briefly, we fitted all possible models containing simpler combinations of fixed effect predictors and ranked them for model fit based on AIC [64]. We only fitted and AIC-ranked models that included a set of predictors hypothesized to have a biological effect on a focal response variable. With this approach, the best-supported model is the one with the lowest AIC value. ΔAIC values were then calculated for every model as the difference between the AIC of the focal model and that of the best-supported model. We report our results in the main text following the full-model approach and LRT outline above and provide AIC model selection tables in S4, S5, S8, S9, S10, and S11 Tables (including models with ΔAIC < 6). Both approaches generate similar results and lead to the same conclusions. Our interpretation of the findings is robust to the choice of statistical framework. When interactive terms (e.g., quadratic terms) were included in a given model, the constituent single terms were always present. Model coefficients are reported and shown in their link-function scale, and models were fitted using maximum likelihood. We formally tested for differences in the effects of male and female helper numbers (i.e., testing whether the slopes of these 2 predictors are statistically different) using Wald $\chi^2$ tests implemented in the R package "car" (v3.1.0; [65]) via its "linearHypothesis" function. Normality and homoscedasticity of model residuals were inspected visually in all models. Statistical analyses were performed in R version 4.2.1. [66], and (unless otherwise specified) statistical models were fitted using the R package "lme4" (v1.1.29; [67]).

**General statistical procedures: Partitioning among-mother and within-mother effects of helpers.** A common concern in studies of the effects of helper numbers on fitness-related traits in cooperative species is that positive correlations between the two could arise not from a causal effect of helpers on the focal trait but instead from both helper numbers and the focal trait being positively impacted by territory (and/or maternal) quality [68,69]. We addressed this concern in 2 ways. First, we excluded young individuals (<6 six months old) from our calculations of the number of male and female helpers (see above; as they contribute little to helping), given that transient resource peaks could leave recent and current productivity positively correlated, potentially yielding a spurious correlation between helper number and current productivity if recently fledged young were considered helpers. Second, we first carried out our analyses using the number of (male and female) helpers as the focal predictor and then partitioned this variable into its within- and among-mother components: Δ (male or female) helper number and μ (male or female) helper number, respectively [32]. "μ helper number" is the mean helper number that a mother had across all of her breeding attempts in the relevant dataset, whereas "Δ helper number" is the difference between her helper number in the focal clutch or brood and "μ helper number." This approach allows us to statistically isolate the effects of within-mother (Δ) variation in helper number (which is both within-mother and within-territory, as each mother in our analyses only ever held one territory), which are indicative of maternal plasticity, in the knowledge that its effects cannot be attributed to variation in quality among mothers or their territories. A recent study has shown that partitioning within- and among-individual effects following this approach provides a robust estimation of the within-individual effect size [43]; the parameter of most interest in this study, as a significant effect of within-mother variation in helper number, would be indicative of a plastic maternal response. As this method can produce biased estimates of the among-individual effect under some circumstances, we have not formally tested for statistical differences between the estimated slopes of the within- and among-individual effects. We also confirmed that a second, more data

demanding, method for estimating within-individual slopes (a bivariate modeling approach) yielded similar effect size estimates for the within-mother slopes (S1 File, Section H).

## Supporting information

**S1 File.** Supporting information include the following: (A) Identification of the time windows of effect for temperature and rainfall. (B) Contextualizing the reported effect size of female helper number on egg volume. (C) No evidence of maternal adjustment of number of clutches laid per year according to helper numbers. (D) Maternal plasticity in egg volume cannot be readily attributed to carryover effects of past help. (E) No evidence of carry-over effects of past help on maternal body condition at laying. (F) Female helper effects on egg volume do not depend on time since the last breeding attempt or the number of helpers. (G) Variation in egg volume is better explained by "current" number of helpers than by number of helpers in the previous breeding attempt. (H) Bivariate models to estimate within- and among-female effects on egg volume and maternal provisioning rates. (I) Seeking evidence of a trade-off between egg volume and maternal provisioning rate. (J) Verifying maternal plasticity in egg size to the availability of female helpers when egg position data is missing.
(PDF)

**S1 Fig. Number of helpers at laying predicts the number of helpers during the nestling rearing period.** Mean ± standard deviation (SD) is presented for both male and female helper numbers (dashed line indicates a 1:1 relationship). For female helper number, linear model: $N = 271$ breeding attempts, $\beta = 0.94 \pm 0.017$. For male helper number, linear model: $N = 271$ breeding attempts, $\beta = 0.93 \pm 0.022$). The data underlying this figure can be found at https://doi.org/10.5281/zenodo.8385995.
(TIF)

**S2 Fig. Sliding window analysis for the effect "heat waves" (days above 35˚C) on egg volume.** See Section A in S1 File above for methods and interpretation. (**a**) Effect of the best-supported "heat waves" index (i.e., that calculated for 0–13 days prior to egg laying) on egg volume when tested within the baseline model. Raw data points in black and regression line (± SE) in blue. (**b**) AIC support (i.e., the difference in AIC between a given sliding window model and the baseline model) for all possible sliding windows of >4 days in length within the 80 days before egg laying. The darker the color of the tiles, the stronger the support for a given window. (**c**) Histogram showing the AIC support for the best-supported heat wave index windows arising from 25 randomisations (i.e., the distribution of AIC support expected if no relationship exists between the heat waves index and egg volume). The blue dashed line illustrates the AIC support achieved using the best-supported window from the real data set. The data underlying this figure can be found at https://doi.org/10.5281/zenodo.8385995.
(TIF)

**S3 Fig. Sliding window analysis for the effect of total rainfall (mm) on egg volume.** See Section A in S1 File above for methods and interpretation. (**a**) Effect of the best-supported total rainfall index (i.e., that calculated for 49–44 days prior to egg laying) on egg volume when tested within the baseline model. Raw data points in black and regression line (± SE) in blue. (**b**) AIC support (i.e., difference in AIC between a given sliding window model and the baseline model) for all possible sliding windows of >4 days in length within the 80 days before egg laying. The darker the color of the tiles, the stronger the support for a given window. (**c**) Histogram showing the AIC support for the best-supported rainfall index windows arising from the 25 randomizations (i.e., the distribution of AIC support expected if no relationship exists between the rainfall index and egg volume). The blue dashed line illustrates the AIC support

achieved using the best-supported window from the real data set. The data underlying this figure can be found at https://doi.org/10.5281/zenodo.8385995.
(TIF)

**S4 Fig. Sliding window analysis for the effect of "heat waves" (days above 35˚C) on maternal provisioning rate.** See Section A in S1 File above for methods and interpretation. (**a**) Effect of the best-supported "heat waves" index (i.e., that calculated for 59–51 days prior to egg laying) on maternal provisioning rate when tested within the baseline model. Raw data points in black and regression line (± SE) in blue. (**b**) AIC support (i.e., difference in AIC between a given sliding window model and the baseline model) for all possible sliding windows of >4 days in length within the 80 days before egg laying. The darker the color of the tiles, the stronger the support for a given window. (**c**) Histogram showing the AIC support for the best-supported heat waves index windows arising from 25 randomizations (i.e., the distribution of AIC support expected if no relationship exists between the heat waves index and maternal provisioning rate). The blue dashed line illustrates the AIC support achieved using the best-supported window from the real data set. The data underlying this figure can be found at https://doi.org/10.5281/zenodo.8385995.
(TIF)

**S5 Fig. Sliding window analysis for the effect of total rainfall (mm) on maternal provisioning rate.** See Section A in S1 File above for methods and interpretation. (**a**) Effect of the best-supported total rainfall index (i.e., that calculated for 78–61 days prior to egg laying) on maternal provisioning rate when tested within the baseline model. Raw data points in black and regression line (± SE) in blue. (**b**) AIC support (i.e., difference in AIC between a given sliding window model and the baseline model) for all possible sliding windows of >4 days in length within the 80 days before egg laying. The darker the color of the tiles, the stronger the support for a given window. (**c**) Histogram showing the AIC support for the best-supported rainfall index windows from each of the 25 randomizations (i.e., the distribution of AIC support expected if no relationship exists between the rainfall index and maternal provisioning rate). The blue dashed line illustrates the AIC support achieved using the best-supported window from the real data set. The data underlying this figure can be found at https://doi.org/10.5281/zenodo.8385995.
(TIF)

**S6 Fig. Distribution of the number of female and male helpers within and among mothers in the dataset used for egg volume analysis.** Analogous distributions for female and male indicate that the power to detect female and male effects was similar. The data underlying this figure can be found at https://doi.org/10.5281/zenodo.8385995.
(TIF)

**S7 Fig. Distribution of observation of egg volume and provisioning effort per female.** (**a**) Number of eggs per mother included in our egg volume analysis and (**b**) number of broods per female in our maternal provisioning analysis. The data underlying this figure can be found at https://doi.org/10.5281/zenodo.8385995.
(TIF)

**S1 Table. Egg volume (cm$^3$) effects on egg mass (g).** Model estimates, standard errors (SE), and their 95% confidence intervals (CI (95%)) are provided along with results from likelihood-ratio tests ($\chi^2_{df = 1}$ and associated $p$-values) assessing the statistical significance of each predictor within the full model (i.e., a model containing all of the terms in the table below). Random

effect standard deviation: "mother ID" = 0.021 g.
(DOCX)

**S2 Table. Egg volume (cm$^3$) effects on hatchling mass (g).** Model estimates, standard errors (SE), and their 95% confidence intervals (CI (95%)) are provided along with results from likelihood-ratio tests ($\chi^2_{df = 1}$ and associated $p$-values) assessing the statistical significance of each predictor within the full model (i.e., a model containing all of the terms in the table below). Random effect standard deviation: "mother ID" = 0.230 g.
(DOCX)

**S3 Table. Within- (Δ) and among-mother (μ) effects of egg volume (cm$^3$) on hatchling mass (g).** Note that partitioning within-mother (Δ) and among-mother (μ) components might produce bias in the among mother component [70]; however, this method produces robust estimation of the within-mother component, which the evidence of plasticity is based upon. Model estimates, standard errors (SE), and their 95% confidence intervals (CI (95%)) are provided along with results from likelihood-ratio tests ($\chi^2_{df = 1}$ and associated $p$-values) assessing the statistical significance of each predictor within the full model (i.e., a model containing all of the terms in the table below). Random effect standard deviation: "mother ID" = 0.231 g.
(DOCX)

**S4 Table. Model selection table for models explaining variation in egg volume (cm$^3$).** This table presents all models within ΔAIC < 6 of the top model. Model coefficients (effect sizes ± standard errors [SE]) are shown along with number of model parameters ("k"), AIC and ΔAIC. "Heat waves" (days above 35˚C), "Clutch size," and "Egg position" were mean centered and scaled by one standard deviation prior model fit to improve model convergence. Similarly, "Rainfall" and "Rainfall$^2$" were fitted as orthogonal vectors, and their estimates are not back transformed in this table (i.e., units do not refer to the real data scale).
(DOCX)

**S5 Table. Model selection table for models explaining variation in egg volume (cm$^3$), when population-level variation in female and male helper number were partitioned into their within-mother (Δ) and among-mother (μ) components prior to model selection.** The table presents all models within ΔAIC < 6 of the top model. Model coefficients (effect sizes ± standard errors [SE]) are shown along with number of model parameters ("k"), AIC and ΔAIC. "Heat waves" (days above 35˚C), "Clutch size," and "Egg position" were mean centered and scaled by one standard deviation prior model fit to improve model convergence. Similarly, "Rainfall" and "Rainfall$^2$" were fitted as orthogonal vectors, and their estimates are not back transformed in this table (i.e., units do not refer to the real data scale).
(DOCX)

**S6 Table. Summary of results of a generalized linear mixed models with zero-truncated Poisson error explaining variation in clutch size ($N$ = 344 clutches eggs laid by 66 mothers in 37 social groups).** Model estimates, standard errors (SE), and their 95% confidence intervals (CI (95%)) are provided along with results from likelihood-ratio tests ($\chi^2_{df = 1}$ and associated $p$-values) assessing the statistical significance of each predictor within the full model (i.e., a model containing all of the terms in the table below). Random effect standard deviation: "season" = 0 clutches, "group ID" = 0 clutches, "mother ID" = 0 clutches.
(DOCX)

**S7 Table. Summary of results of a generalized linear mixed models with zero-truncated Poisson error explaining variation in clutch size ($N$ = 344 clutches eggs laid by 66 mothers in 37 social groups) when population-level variation in female and male helper number**

were partitioned into their within-mother (Δ) and among-mother (μ) components. Model estimates, standard errors (SE), and their 95% confidence intervals (CI (95%)) are provided along with results from likelihood-ratio tests ($\chi^2_{df = 1}$ and associated $p$-values) assessing the statistical significance of each predictor within the full model (i.e., a model containing all of the terms in the table below). Random effect standard deviation: "season" = 0 clutches, "group ID" = 0 clutches, "mother ID" = 0 clutches.
(DOCX)

**S8 Table. Model selection table for models explaining variation in clutch size (zero-truncated models).** This analysis revealed very week evidence for an effect of female or male helper number on clutch size. Models including female or male helper number received similar support from the data that the intercept-only model. Model coefficients (effect sizes ± standard errors) are shown along with number of model parameters ("k"), AIC and ΔAIC.
(DOCX)

**S9 Table. Model selection table for models explaining variation in clutch size (zero-truncated models) after partitioning variation in female and male helper number into their within-mother (Δ) and among-mother (μ) components.** Models including female or male helper number components received similar support from the data that the intercept-only model. Model coefficients (effect sizes ± standard errors) are shown along with number of model parameters ("k"), AIC and ΔAIC. "Clutch order" was mean centered and scaled by one standard deviation prior model fit to improve model convergence.
(DOCX)

**S10 Table. Model selection table for models explaining variation in maternal provisioning rate (feeds/hour).** This table presents all models within ΔAIC < 6 of the top model. Model coefficients (effect sizes ± standard errors [SE]) are shown along with number of model parameters ("k"), AIC and ΔAIC. "Heat waves" (days above 35˚C) and "Brood size" were mean centered and scaled by one standard deviation prior model fit to improve model convergence. Similarly, "Rainfall" and "Rainfall$^2$" were fitted as orthogonal vectors, and their estimates are not back transformed in this table (i.e., units do not refer to the real data scale).
(DOCX)

**S11 Table. Model selection table for models explaining variation in maternal provisioning rate (feeds/hour), when population-level variation in female and male helper number were partitioned into their within-mother (Δ) and among-mother (μ) components prior to model selection.** Model coefficients (effect sizes ± standard errors) are shown along with number of model parameters ("k"), AIC and ΔAIC. "Heat waves" (days above 35˚C) and "Brood size" were mean centered and scaled by one standard deviation prior model fit to improve model convergence. Similarly, "Rainfall" and "Rainfall$^2$" were fitted as orthogonal vectors, and their estimates are not back transformed in this table (i.e., units do not refer to the real data scale).
(DOCX)

**S12 Table. Summary of results of a linear mixed model explaining variation in egg volume (cm$^3$), including every main effect of interest, but including data for rainfall values below its peak ($N$ = 466 eggs).** Model estimates, standard errors (SE), and their 95% confidence intervals (CI (95%)) are provided along with results from likelihood-ratio tests ($\chi^2_{df = 1}$ and associated $p$-values) assessing the statistical significance of each predictor within the full model. "Rainfall," "Heat waves" (days above 35˚C), "Clutch size," and "Egg position" were mean centered and scaled by one standard deviation prior model fit to improve model convergence.
(DOCX)

**S13 Table. Summary of results of a linear mixed model explaining variation in egg volume (cm³), including every main effect of interest after population-level variation in female and male helper number were partitioned into their within-mother (Δ) and among-mother (μ) components, including data for rainfall values below its peak (*N* = 466 eggs).** Model estimates, standard errors (SE), and their 95% confidence intervals (CI (95%)) are provided along with results from likelihood-ratio tests ($\chi^2_{df = 1}$ and associated *p*-values) assessing the statistical significance of each predictor within the full model. "Rainfall," "Heat waves" (days above 35˚C), "Clutch size," and "Egg position" were mean centered and scaled by one standard deviation prior model fit to improve model convergence.
(DOCX)

**S14 Table. Summary of results of a linear mixed model explaining variation in egg volume (cm³), without the inclusion of rainfall and at the population level (*N* = 490 eggs).** Model estimates, standard errors (SE), and their 95% confidence intervals (CI (95%)) are provided along with results from likelihood-ratio tests ($\chi^2_{df = 1}$ and associated *p*-values) assessing the statistical significance of each predictor within the full model. "Heat waves" (days above 35˚C), "Clutch size," and "Egg position" were mean centered and scaled by one standard deviation prior model fit to improve model convergence. "df" = degrees of freedom for likelihood-ratio tests.
(DOCX)

**S15 Table. Summary of results of a linear mixed model explaining variation in egg volume (cm³), without the inclusion of rainfall and after population-level variation in female and male helper number were partitioned into their within-mother (Δ) and among-mother (μ) components (*N* = 490 eggs).** Model estimates, standard errors (SE), and their 95% confidence intervals (CI (95%)) are provided along with results from likelihood-ratio tests ($\chi^2_{df = 1}$ and associated *p*-values) assessing the statistical significance of each predictor within the full model. "Heat waves" (days above 35˚C), "Clutch size," and "Egg position" were mean centered and scaled by one standard deviation prior model fit to improve model convergence.
(DOCX)

**S16 Table. Summary of results of a linear mixed model explaining variation in maternal provisioning rate (feeds/hour), without the inclusion of rainfall and at the population level.** Model estimates, standard errors (SE), and their 95% confidence intervals (CI (95%)) are provided along with results from likelihood-ratio tests ($\chi^2_{df = 1}$ and associated *p*-values) assessing the statistical significance of each predictor within the full model. "Heat waves" (days above 35˚C) and "Brood size" were mean centered and scaled by one standard deviation prior model fit to improve model convergence.
(DOCX)

**S17 Table. Summary of results of a linear mixed model explaining variation in maternal provisioning rate (feeds/hour), without the inclusion of rainfall and after population-level variation in female and male helper number were partitioned into their within-mother (Δ) and among-mother (μ) components.** Model estimates, standard errors (SE), and their 95% confidence intervals (CI (95%)) are provided along with results from likelihood-ratio tests ($\chi^2_{df = 1}$ and associated *p*-values) assessing the statistical significance of each predictor within the full model. "Heat waves" (days above 35˚C) and "Brood size" were mean centered and scaled by one standard deviation prior model fit to improve model convergence.
(DOCX)

**S18 Table. Summary of results of a generalized linear mixed model (Poisson error structure) explaining variation in the number of clutches laid per year.** Model estimates, standard errors (SE), and their 95% confidence intervals (CI (95%)) are provided along with results from likelihood-ratio tests ($\chi^2_{df = 1}$ and associated $p$-values) assessing the statistical significance of each predictor within the full model. Random effect standard deviation: "mother ID" = 0 clutches, "group ID" = 0 clutches; breeding season = 0.470 clutches ($\chi^2_1 = 64.65$, $p < 0.001$).
(DOCX)

**S19 Table. Summary of results of a generalized linear mixed model (Poisson error structure) explaining variation in the number of clutches laid per year after population-level variation in female and male helper number were partitioned into their within-mother ($\Delta$) and among-mother ($\mu$) components.** Model estimates, standard errors (SE), and their 95% confidence intervals (CI (95%)) are provided along with results from likelihood-ratio tests ($\chi^2_{df = 1}$ and associated $p$-values) assessing the statistical significance of each predictor within the full model. Random effect standard deviation: "mother ID" = 0 clutches, "group ID" = 0 clutches; breeding season = 0.45 clutches ($\chi^2_1 = 55.81$, $p < 0.001$).
(DOCX)

**S20 Table. Effect of the number of female and male helper numbers on a previous breeding attempt on egg volume (cm$^3$).** Model estimates, standard errors (SE), and their 95% confidence intervals (CI (95%)) are provided along with results from likelihood-ratio tests ($\chi^2_{df = 1}$ and associated $p$-values) assessing the statistical significance of each predictor within the full model.
(DOCX)

**S21 Table. Effect of time since the last breeding attempt on egg volume (cm$^3$).** Model estimates, standard errors (SE), and their 95% confidence intervals (CI (95%)) are provided along with results from likelihood-ratio tests ($\chi^2_{df = 1}$ and associated $p$-values) assessing the statistical significance of each predictor within the full model.
(DOCX)

**S22 Table. Bivariate model of egg volume (cm$^3$) and number of female helpers.** This model was fitted in R using the MCMCglmm package (v2.34; [71]), using an inverse Gamma as prior distribution for random and residual (co)variances (V = 1, nu = 1.002), 50,000 MCMC iterations, with 1,000 as initial burn-in and sampling every 10 iterations. Effective sample sizes for all model terms were always higher than 4,000, and MCMC traces were visually inspected to confirm convergence. We calculated the slope between egg volume and female helper number within and among mothers by dividing the corresponding covariance between egg volume and female helper number by the estimated variance in female helper number. We found that both the within- and among-mother slopes of female helper number on egg volume were positive (within mothers [95% CrI] = 0.015 [−0.004, 0.033]; among mothers [95% CrI] = 0.041 [−0.126, 0.214]). Mean posterior estimates ("Mean") and their 95% credible intervals ("95% CI") are provided.
(DOCX)

**S23 Table. Bivariate model of maternal provisioning rates (feeds/hour) and number of female helpers.** This model was fitted in R using the MCMCglmm package (v2.34; [71]), using an inverse Gamma as prior distribution for random and residual (co)variances (V = 1, nu = 1.002), 50,000 MCMC iterations, with 1,000 as initial burn-in and sampling every 10 iterations. Effective sample sizes for all model terms were always higher than 2,500, and MCMC

traces were visually inspected to confirm convergence. We calculated the slope between maternal provisioning rates and female helper number within and among mothers by dividing the corresponding covariance between maternal provisioning rates and female helper number by the estimated variance in female helper number. We found that both the within- and among-mother slopes of female helper number on maternal provisioning rates were negative (within mothers [95% CrI] = −0.451 [−1.077, 0.151]; among mothers [95% CrI] = −0.311 [−1.381, 0.667]). Mean posterior estimates ("Mean") and their 95% credible intervals ("95% CI") are provided.
(DOCX)

**S24 Table. Bivariate model of egg volume (cm$^3$) and maternal provisioning rate (feeds/hour) not including helper numbers as predictors.** This analysis used the full data set from the egg volume model in Table 1 (490 measures from 271 clutches) and the maternal provisioning rate model in Table 3 (124 measures from 124 broods). The model included 59 clutches/broods that had measures for both egg volume and provisioning rate, while 277 clutches/broods had measures for only one of these 2 metrics. This model was fitted in R using the brms package (v2.34; [72]), using a default prior distribution for random variance components and a normal distribution of mean 0 and standard deviation 100 for fixed effects, and 4 chains of 50,000 MCMC iterations, with 25,000 as initial burn-in and sampling every 10 iterations in each case. Residual variation for maternal provisioning rates was fixed to 0.01 (i.e., no residual variation was left after accounting for differences among clutches). Convergence was assessed via Rhat values, which were always below 1.01 and visual inspection of chain traces. Response terms and fixed effect variables were mean centered and scaled by one standard deviation prior model fit to improve model convergence. "Rainfall" and "Rainfall$^2$" were fitted as orthogonal polynomial, and their estimates are not back transformed in this table (i.e., units do not refer to the real data scale). Mean posterior estimates ("Mean") and their 95% credible intervals ("95% CI") are provided.
(DOCX)

**S25 Table. Bivariate model of egg volume (cm$^3$) and maternal provisioning rates (feeds/hour) including the number of helpers as predictors.** This analysis used the full data sets from the egg volume model in Table 1 (490 measures from 271 clutches) and the maternal provisioning rate model in Table 3 (124 measures from 124 broods). Approximately 59 clutches/broods had measures for both egg volume and provisioning rate, while 277 clutches/broods had measures for only one of these 2 metrics. This model was fitted in R using the brms package (v2.34; [72]), using a default prior distribution for random variance components and a normal distribution of mean 0 and standard deviation 100 for fixed effects, and 4 chains of 50,000 MCMC iterations, with 25,000 as initial burn-in and sampling every 10 iterations in each case. Residual variation for maternal provisioning rates was fixed to 0.01 (i.e., no residual variation was left after accounting for differences among clutches). Convergence was assessed via Rhat values, which were always below 1.01 and visual inspection of chain traces. Response terms and fixed effect variables were mean centered and scaled by one standard deviation prior model fit to improve model convergence. "Rainfall" and "Rainfall$^2$" were fitted as orthogonal polynomial, and their estimates are not back transformed in this table (i.e., units do not refer to the real data scale). Mean posterior estimates ("Mean") and their 95% credible intervals ("95% CI") are provided.
(DOCX)

**S26 Table. Summary of results of a linear mixed model explaining variation in egg volume (cm$^3$) using the extended dataset that includes low-quality observations not containing**

**information on egg position within the clutch ($N$ = 906 eggs).** This model contains data for a single clutch of 4 eggs. Model estimates, standard errors (SE), and their 95% confidence intervals (CI (95%)) are provided along with results from likelihood-ratio tests ($\chi^2_{df\,=\,1}$ and associated $p$-values) assessing the statistical significance of each predictor within the full model (i.e., a model containing all of the terms in the table below). Random effect standard deviation: "season" = 0.001 cm$^3$, "group ID" = 0 cm$^3$, "clutch ID" = 0.014 cm$^3$, "mother ID" = 0.078 cm$^3$. "Heat waves" (days above 35˚C) and "Clutch size" were mean centered and scaled by one standard deviation prior model fit to improve model convergence. "Rainfall" and "Rainfall$^2$" were fitted as orthogonal polynomial, and their estimates are not back transformed in this table (i.e., units do not refer to the real data scale).
(DOCX)

**S27 Table. Summary of results of a linear mixed model explaining variation in egg volume (cm$^3$) using the extended dataset that includes low-quality observations not containing information on egg position within the clutch and only clutches of one egg ($N$ = 35 eggs).** Model estimates, standard errors (SE), and their 95% confidence intervals (CI (95%)) are provided along with results from likelihood-ratio tests ($\chi^2_{df\,=\,1}$ and associated $p$-values) assessing the statistical significance of each predictor within the full model (i.e., a model containing all of the terms in the table below). Random effect standard deviation: "season" = 0 cm$^3$, "group ID" = 0.197 cm$^3$, "mother ID" = 0 cm$^3$. "Heat waves" (days above 35˚C) and "Clutch size" were mean centered and scaled by one standard deviation prior model fit to improve model convergence. "Rainfall" and "Rainfall$^2$" were fitted as orthogonal polynomial, and their estimates are not back transformed in this table (i.e., units do not refer to the real data scale).
(DOCX)

**S28 Table. Summary of results of a linear mixed model explaining variation in egg volume (cm$^3$) using the extended dataset that includes low-quality observations not containing information on egg position within the clutch and only clutches of three eggs ($N$ = 184 eggs).** Model estimates, standard errors (SE), and their 95% confidence intervals (CI (95%)) are provided along with results from likelihood-ratio tests ($\chi^2_{df\,=\,1}$ and associated $p$-values) assessing the statistical significance of each predictor within the full model (i.e., a model containing all of the terms in the table below). Random effect standard deviation: "season" = 0 cm$^3$, "group ID" = 0 cm$^3$, "clutch ID" = 0.001 cm$^3$, "mother ID" = 0.080 cm$^3$. "Heat waves" (days above 35˚C) and "Clutch size" were mean centered and scaled by one standard deviation prior model fit to improve model convergence. "Rainfall" and "Rainfall$^2$" were fitted as orthogonal polynomial, and their estimates are not back transformed in this table (i.e., units do not refer to the real data scale).
(DOCX)

**S29 Table. Summary of results of a linear mixed model explaining variation in egg volume (cm$^3$) using the extended dataset that includes low-quality observations not containing information on egg position within the clutch and only clutches of 2 eggs ($N$ = 683 eggs).** Model estimates, standard errors (SE), and their 95% confidence intervals (CI (95%)) are provided along with results from likelihood-ratio tests ($\chi^2_{df\,=\,1}$ and associated $p$-values) assessing the statistical significance of each predictor within the full model (i.e., a model containing all of the terms in the table below). Random effect standard deviation: "season" = 0.001 cm$^3$, "group ID" = 0 cm$^3$, "clutch ID" = 0.019 cm$^3$, "mother ID" = 0.078 cm$^3$. "Heat waves" (days above 35˚C) and "Clutch size" were mean centered and scaled by one standard deviation prior model fit to improve model convergence. "Rainfall" and "Rainfall$^2$" were fitted as orthogonal polynomial, and their estimates are not back transformed in this table (i.e., units do not refer

to the real data scale).
(DOCX)

**S30 Table. Summary of results of a linear mixed model explaining variation in egg volume (cm$^3$) using the extended dataset that includes low-quality observations not containing information on egg position within the clutch and including interactions between female and male helper numbers and clutch size as a categorical (i.e., factor) variable ($N$ = 906 number of eggs).** This model contains data for a single clutch of 4 eggs. Model estimates, standard errors (SE), and their 95% confidence intervals (CI (95%)) are provided along with results from likelihood-ratio tests ($\chi^2_{df = 1}$ and associated $p$-values) assessing the statistical significance of each predictor within the full model (i.e., a model containing all of the terms in the table below). Random effect standard deviation: "season" = 0 cm$^3$, "group ID" = 0 cm$^3$, "clutch ID" = 0.013 cm$^3$, "mother ID" = 0.078 cm$^3$. "Heat waves" (days above 35˚C) and "Clutch size" were mean centered and scaled by one standard deviation prior model fit to improve model convergence. "Rainfall" and "Rainfall$^2$" were fitted as orthogonal polynomial, and their estimates are not back transformed in this table (i.e., units do not refer to the real data scale).
(DOCX)

**S31 Table. Summary of results of a generalized linear mixed model (Poisson residual distribution) explaining variation in maternal provisioning feeds, including log observation time as an offset and, thus, effectively modeling maternal provisioning rate (feeds/hour; $N$ = 251 days of maternal provisioning observations).** Model estimates, standard errors (SE), and their 95% confidence intervals (CI (95%)) are provided in the link scale (i.e., "log") along with results from likelihood-ratio tests ($\chi^2_{df = 1}$ and associated $p$-values) assessing the statistical significance of each predictor within the full model (i.e., a model containing all of the terms in the table below). Random effect standard deviation: "season" = 0.090 feeds/hour, "group ID" = 0 feeds/hour, "clutch ID" = 0.240 feeds/hour, "mother ID" = 0.084 feeds/hour, observation level = 0.341 feeds/hour. "Heat waves" (days above 35˚C) and "Brood size" were mean centered and scaled by one standard deviation prior model fit to improve model convergence. "Rainfall" and "Rainfall$^2$" were fitted as orthogonal polynomial, and their estimates are not back transformed in this table (i.e., units do not refer to the real data scale).
(DOCX)

## Acknowledgments

We would like to thank the many team members who contributed to the collection of the long-term data over the years (in particular Tom Reed, Jenny York, Dom Cram, Lindsay Walker, Emma Wood, and Xavier Harrison), Northern Cape Conservation for permission to carry out the research, Nigel Bennett for invaluable assistance with in-country permissions, and E. Oppenheimer & Son, the Tswalu Foundation, and all at Tswalu Kalahari Reserve for their support in the field and the collection and sharing of the reserve-wide rainfall data. We also thank Ben Hatchwell and Erik Postma for insightful discussions.

## Author Contributions

**Conceptualization:** Pablo Capilla-Lasheras, Andrew J. Young.

**Data curation:** Pablo Capilla-Lasheras, Andrew J. Young.

**Formal analysis:** Pablo Capilla-Lasheras, Alastair J. Wilson.

**Funding acquisition:** Andrew J. Young.

**Investigation:** Pablo Capilla-Lasheras, Alastair J. Wilson, Andrew J. Young.

**Methodology:** Pablo Capilla-Lasheras, Andrew J. Young.

**Project administration:** Andrew J. Young.

**Resources:** Andrew J. Young.

**Supervision:** Alastair J. Wilson, Andrew J. Young.

**Visualization:** Pablo Capilla-Lasheras.

**Writing – original draft:** Pablo Capilla-Lasheras.

**Writing – review & editing:** Pablo Capilla-Lasheras, Alastair J. Wilson, Andrew J. Young.

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
