## [Editor Report · Decision Letter 0]

22 Dec 2022

Dear Dr Capilla-Lasheras, 

Thank you for submitting a new version of your manuscript entitled "Mothers front-load their investment to the egg stage when helped in a wild cooperative bird" for consideration as a Research Article by PLOS Biology.

Your revisions have now been evaluated by the PLOS Biology editorial staff, and I'm writing to let you know that we would like to send your submission out for re-review.

However, before we can send your manuscript back to reviewers, we need you to complete your submission by providing the metadata that is required for full assessment. To this end, please login to Editorial Manager where you will find the paper in the 'Submissions Needing Revisions' folder on your homepage. Please click 'Revise Submission' from the Action Links and complete all additional questions in the submission questionnaire.

Once your full submission is complete, your paper will undergo a series of checks in preparation for re-review. After your manuscript has passed the checks it will be sent out for review. To provide the metadata for your submission, please Login to Editorial Manager (https://www.editorialmanager.com/pbiology) within two working days, i.e. by Dec 24 2022 11:59PM.

Kind regards,

Roli Roberts

Roland Roberts, PhD

Senior Editor

PLOS Biology

rroberts@plos.org

---

## [Decision Letter · Decision Letter 1]

21 Feb 2023

Dear Dr Capilla-Lasheras,

Thank you for your patience while we considered your revised manuscript "Mothers front-load their investment to the egg stage when helped in a wild cooperative bird" for publication as a Research Article at PLOS Biology. Your revised study has been evaluated by the PLOS Biology editors, the Academic Editor, and two of the original reviewers.

You'll see that reviewer #1 now thinks that the paper is much clearer, but raises concerns about some aspects of your interpretation, and has a large number of residual questions about key aspects of the methodology. Reviewer #3 also sees a significant improvement in the paper, but still pushes for several analyses that he either requested last time, or which he feels have been done inappropriately.

In light of the reviews, which you will find at the end of this email, we would like to invite you to revise the work to thoroughly address the reviewers' reports.

Given the extent of revision needed, we cannot make a decision about publication until we have seen the revised manuscript and your response to the reviewers' comments. Your revised manuscript is likely to be sent for further evaluation by all or a subset of the reviewers.

**IMPORTANT - SUBMITTING YOUR REVISION**

*Re-submission Checklist*

*Published Peer Review*

*PLOS Data Policy*

Sincerely,

Roli Roberts

Roland Roberts, PhD

Senior Editor

PLOS Biology

rroberts@plos.org

REVIEWERS' COMMENTS:

Reviewer #1: 

This version of the manuscript has been rendered much more intelligible with the abandonment (or relegation to the SI) of the convoluted AIC methods used in the earlier version. Hence it is now possible to address what has actually been done. The authors contend that females in cooperatively breeding birds can vary egg provisioning in two very different ways in responses to changes in helper number. In one scenario, females can improve their own survival by reducing the investment in eggs in anticipation that the resultant small young will not suffer because larger groups have a dependably greater corporate feeding rate. The alternative hypothesis, which the authors use to explain data from arid-zone sparrow-weavers, is that because the adult females can anticipate greater care from helpers but cannot gain assistance from helpers during the incubation phase, they invest more heavily in eggs and hence can lessen their own contribution to offspring provisioning. 

Indeed, one of the strange things about this system is the massive variation in egg size - the largest eggs are 1.61 cm3 (57%) larger than the smallest eggs. I presume this variation partly motivated the study, as such variation in a low-fecundity small passerine seems quite unusual. Much of the (presumably) between brood effects are driven in understandable ways by rainfall (0.31cm3 per 100m of rainfall). Within brood effects are driven mainly by egg position (-0.04cm3) from the first to the second egg, though the Methods for this are not given in the paper (see below). The data for within female change in helper number are largely confined to a maximum increase or decrease of 3 from the average produced by the female, which is predicted to lead to a change of 0.057cm3, or about 3.5% of the natural variation in egg size. A change in one helper is therefore about 1% of the natural variation (as can be understood by looking at the cloud of points in Fig. 1b). I am afraid I still find the likely biological effect of this uncompelling. The appropriate data are not really presented in this manuscript, but the authors suggest that heavier new-hatched chicks (unsurprisingly from heavier eggs) are more likely to thrive, yet the effects of helper number seem to be more profound for post-natal provisioning rates. However, this difference is difficult to interpret in the absence of corporate provisioning rates. What is clear is that the mother responds far more dramatically to variation in brood size than she does to the work force available. However, we don't know if similar responses occur in all group members.

A further problem is that although the authors present the among female variation as a solution to the pervasive problem of whether it is possible to attribute the direction of causation in correlations between performance and helper number in cooperatively breeding societies. This method has been extensively discussed by Dickinson & Hatchwell (2004) Chapter 3 in Ecology and Evolution of Cooperatively Breeding in Birds, and I do not think that the difficulties they discuss are addressed by this method. In the present case, there is no help given to the reader concerning why the number of helpers fluctuate from clutch to clutch. The expectations are quite different if either: i) as is plausible in a desert environment, the reduction in helpers is associated with a collapse in population number, or ii. if amelioration in conditions through increased rainfall and the absence of heatwaves is associated with splitting of groups as more birds are able to attempt independent reproduction. 

The authors correctly point out that conclusions can be drawn from a greater change in numbers of female helpers than an equivalent change in the number of male helpers. However, it is not really explained why this is the case. Indeed, since Cockburn (1198, ARES) drew attention to the paradox that male-biased help is more common in cooperatively breeding birds, but female helpers more often seem to increase reproductive success, there has been no real progress in understanding of why this is the case.

Technical comments

Methods

436-441 - I could not find methods for converting volume to mass - was this done with Hoyt's formula. If so, it should be stated.

Results

189-191. How much of maternal repeatability is associated with between versus within brood effects (see lines 184-196).

191-194. How were individual eggs marked and then connected to individual chicks, as is done on Fig 3. I couldn't find this anywhere in the paper, though 'egg position' is used as an important variable in the models, and is highly significant in Table 1. It is quite unusual to have substantial hatching asynchrony in birds with a clutch size of 2. If it assumed that the first hatched chick is associated with the first laid egg the pattern may exist through the assumption rather than any biological effect.

200-201. This is an important point, and it is strange that it demonstrated by reference to Supplementary Information in another paper.

238-251. It seems strange that there is no effect of clutch size on egg volume, but there is a very strong effect of position in the clutch.

255+. The difference between male and female helpers raises the question of whether dominant males and other helpers also load-lighten, but this cannot be assessed.

Reviewer #3:

[identifies himself as Joel Pick]

I think this version of the manuscript is much improved from the last one. Many of my previous points have been addressed by the authors adoption of a different analytical approach. It is nice to see that this doesn't substantially affect the results, and some interesting patterns remain. 

There are still a few points which I do not feel have been addressed sufficiently in the response and revised manuscript and remain unresolved. I don't think that they will alter the main message of the paper (the within mother effects of helper number on egg size and provisioning rate), but I think it is worth using the appropriate statistical models and trying to present unbiased results where possible.

The main comment relate to (addressed in more detail below)

 - explicit testing for a within mother relationship between egg size and provisioning rate (comment 1)

 - use of multivariate models to reduce bias in estimates (comment 2)

 - why the egg size data set is halved, based on whether data on laying order is available (comment 20)

In reply to my previous comments and the responses:

1. Testing for a relationship between egg size and provisioning rate (my previous comment 3).

In the response to my previous comment, it is argued that this wouldn't be meaningful without an experiment. But the same can be said for the relationship between maternal expenditure and helper number. This is observational data and has its limitations, but I don't see why that is reason to not fully explore the hypothesis suggested here. For example, if female helper number provides general benefits to the group, then group size could reflect maternal condition pre-laying, and so the increase in egg size could be a function of maternal condition, rather than anticipation of future care from female helper. To me this seems completely plausible, but is not an alternative explanation explored in the manuscript, and cannot be ruled out without an experiment. Therefore, I think adopting a within/between mother approach to the relationships between egg size and provisioning rate would be no less reasonable than the analyses presented in the manuscript, and directly test the prediction that a change in egg size is accompanied by a change in provisioning rate, mediated by helper number (L 172).

A nice approach to showing that any relationship between the two was mediated by helper number would be to run a bivariate model of egg size and provisioning rate, within and without helper number as a covariate for both traits, and then see how the within mother covariance changes as a result. This would be similar to adopting a hybrid trait-based/variance component approach used to identify traits mediating maternal effects in quantitative genetics (Mcadam, Garant and Wilson 2014 in Quantitative Genetics in the Wild). Additionally, by having temporal random effects (such as breeding season) in this analysis would allow the problem of a correlation driven by the environment that is raised in the response to be mitigated.

2. Using bivariate models to account for the known bias in within subject centering models (my previous My comment 4).

The authors are right that the main bias in these models is in the between subject slopes. However, given the that between subject slopes and the difference are reported in the paper, it seems that strange to use a method that is known to be biased for the estimation of these additional parameters. For the analysis of provisioning rate in particular, this bias is likely to be large, ad there is a median of 2 observations per mother. I think that either the between mother slopes and difference should not be reported in the main text because of the known bias (and this bias and lack of presentation explained clearly in the manuscript), or a bivariate approach used. 

I would strongly recommend the use of bivariate models, and the presenting of a more complete and unbiased set of parameters. The fact that the within mother slopes appear to be stronger is interesting in of itself. As stated in the introduction, one of the reasons for using the within subject centering approach is that between mother slopes may obscure the within mother slope of interest. However, if anything, the opposite is found. If helper number had a causal effect on the response alone you would expect the same slope in the two components. This means that the mothers are either not reacting to the absolute number of helpers, or there is an additional correlation in the opposite direction between average maternal expenditure and helper number which is interesting to discuss. The difference between the two is not statistically supported with these models, but its should be noted that the bias would decrease the difference between the two slopes, and so the bivariate approach may give greater granularity over whether there is a difference. To me it seems strange not to want to explore this further. 

In terms of the additional complexity of the bivariate model - I disagree. First, linear mixed models are amazingly robust to non-normality of the data (see for example Schielzeth et al 2020 MEE). Second I believe that the inferences would be *more* meaningful on the log scale. On re-reading the manuscript (before reading the responses) it occurred to me that helper number should anyway be log transformed. It is a count, and so inferences make more sense on a multiplicative scale. Not log transforming will also weaken any relationship between helper number and the response, if they are linear on that scale. The same also holds for provisioning rate (which I previously commented upon). The difficulty with log transformation would come with the presence of 0s - this can be handled by adding 0.5 (Yamamura 1999 Population Ecology), but is not very satisfactory. The data could also be treated as Poisson. It's worth noting that because the within subject centering is done with two variables, a bivariate model wouldn't strictly be appropriate. As shown in Froy et al 2019 Science (p6 of supplements) you can extend this bivariate approach in Phillimore et al 2010 PNAS to many variables.

The response also states that a 'bivariate model is more demanding of the dataset'. My reading of this is less likely to give a significant p-value - personally I would advocate being more worried about biased estimates and less about significance thresholds. 

As a slight aside, the response and revised manuscript cites Westneat et al 2020 (citation 59) as a justification for the use of these models. A more appropriate reference is Ludtke et al 2008 Psychological Methods, as a) they derive the biases in these models and b) Westnest et al specifically addresses the scenario where there is between subject variation in the response to the environment (i.e. random slopes), and addresses how models with and without mean centering (but specifying random slopes) cope with matching or mismatching data generating processes. Westnest et al do compare a bivariate model to these random slope models, as bivariate models have been suggested as a solution (Phillimore et al. 2010 PNAS) to the known bias in within subject centering models shown in Ludtke et al 2008, but never use the model without random slopes that is used in this manuscript. 

3. Egg volume and egg mass

I am still unsure why there an analysis of within and between mother effects of egg volume on egg mass. My understanding of this is to show that egg volume is a good proxy for egg mass, which is a better measure of expenditure. In the response to this comment in my previous review the authors simply justify why this has been done for chick mass. The reason for analysis with check mass is clear (indeed I have published a paper promoting this approach for this question), but this doesn't answer what relevance this has to egg volume and egg mass. I think a simple correlation would suffice here.

Also please present the full results of these models in tables (in supplements is fine). The difference between within and among mother effects of egg size on chick mass are important for inferring causality (see Pick et al 2016 Am Nat).

4. Analysis of provisioning rate

The suggestion to log transform provisioning rate and properly adjust for observation period is based on trying model the underlying process, rather than achieving normality of the residuals. This suggestion was based on this preprint (https://ecoevorxiv.org/repository/view/4531/), an updated version of which is in publication. The log transformation would be more meaningful, and I suspect would lead to stronger associations. As stated above, this transformation would make sense for both helper number and provisioning rate. 

Other Comments:

5. L55: None of references 4-8 refer to theory papers, so it seems a bit strange to me to base evolutionary predictions on them.

6. L57 (and throughout): This might be personal preference, but 'per-offspring pre-natal maternal investment' seems a more natural way to phrase this, than 'maternal pre-natal investment per offspring'.

7. With regards to the use of the phrase 'maternal investment' - it might be worth referring to it as maternal expenditure, as it has been argued that maternal investment implies a known cost to the mother and benefit for the offspring, which is not shown here (and typically is very hard to demonstrate). For example it is defined as 'Any investment by the parent in an individual offspring that increases the offspring's survival and reproductive success at the cost of the parent's ability to invest in other current or future offspring' in Royle, Smiseth and Kolliker 2012 Evolution of Parental Care (Table 1.1). Parental expenditure would perhaps be most accurate. Parent care infers a known benefit to the offspring, which (with reference to a comment in my previous review) I don't think is known for wither of these maternal traits.

8. The concept of 'additive post-natal care' seems important to the predictions but is not really explained anywhere. When I first read it I assumed that it meant that each additional helper one average added the same amount of care again (i.e. the effect of helper number was additive), but I think this is wrong. The first reference to additive care (reference 8, Hatchwell 1999) defines it (in the abstract of Hatchwell 1999) as 'When parents maintain the same effort regardless of helper number, helper care is additive'. One justification for the utility of this system is that 'That only female helpers provide demonstrably additive post-natal care provides an unusual opportunity to distinguish the hypothesized pre-natal maternal responses to the availability of additive help' (L150). However, under the definition of Hatchwell 1999, its not known whether the care is additive until the effect on the mother post-natal are is known, and then in this system it would not be, as the mother reduces her care. Perhaps there is more subtlety in the definition than I understand, but it would be worth explaining this concept further.

9. l 115 should be 'a meta-analysis'

10. L 120-134: It seems strange not to reference Fortuna et al 2021 (reference 31), as they did exactly this within and between analysis with helper number in Sociable Weavers. It would be again appropriate to reference it on L143.

11. L 138 'We do so by testing for maternal plasticity in both pre-natal investment per offspring (egg volume; while accounting for effects of clutch size) and post-natal investment per offspring (maternal nestling provisioning rate) according to the availability of help.'

It might be more consistent to write (maternal nestling provisioning rate accounting for brood size), as both clutch size and brood size are accounted for in order to look at per-offspring expenditure.

12. L 200 As in my previous review, the relationship between egg size and chick survival has been previous tested in this system, and no statistical support found. It is disingenuous to cite this paper and suggest otherwise. In the response to my previous comment several possible confounders (such as not accounting for clutch size) were mentioned. From the introduction, it is not completely clear why the analysis on chick size is included. Presumably this is to demonstrate the importance of egg size. It seems to me that a model of chick survival would directly address this issue. If the argument is that it is a power issue, I would think that an analysis based on 490 eggs (or 906 in the full dataset) would be able to predict meaningful effect sizes, or at least a power analysis would show the minimum effect size that was able to be detected, and whether this minimum effect size was meaningful. There is also no analysis looking at the impact of provisioning rate on chick size or survival (as mention in my last review - comment 5), and this is an equally important part of the picture, if stages are being traded off against each other. 

13. L 206 (and throughout) An effect size would be much more useful in the text than a chi squared stat - that and the p-value show the same thing, and the chisquared stats are in the tables. Presenting the effect sizes in the text allows comparison whilst reading, rather than constant reference back to a table (where they are also useful).

14. L214 (and 262) 'A significant difference between the effect sizes for the within- and among-mother components of female helper number would indicate that consistent differences in egg volume among mothers (other than those arising from the maternal plastic response to female helper number) also contribute significantly to the population-level relationship between female helper number and egg volume detected in Table 1'

This is confusing statistical significance with effect size. When the repeatability of helper number is low, the overall effect will be dominated by the within mother effect (as is seen here), and vice versa. A statistically significant difference just shows that the slopes are different, the contribution to the overall effect is to do with the relative amounts of within and between mother variation in the predictor. 

15. Tables - Is rainfall^1 refers to linear effect, I would suggest having rainfall and rainfall^2 instead - this is more standard notation. I was expecting footnotes to the table when I first looked at it. 

16. L 393 ' If, as here, mothers lay larger eggs when helped, controlling for variation in egg size could lead to the underestimation of helper effects on offspring, by factoring out helper effects that arise indirectly via maternal investment in the egg.'

I am not convinced that this is 'dangerous' approach - it simply helps separate direct and indirect effects - you would underestimate the total effects due to helpers, but correctly estimate the direct effects of helpers. It just addresses a different question, and requires researchers to to explicit about the assumptions of their model and what question their model answers. 

17. L 481 It seems strange that measures provisioning rate per brood are averaged over, but not egg size per clutch?

18. L 484 How correlated were successive measures of provisioning rate? 

19. L 524 'These interactive terms are included to specifically test whether the effect of helpers on egg volume is dependent on egg position or clutch size.'

In my last review I mentioned that the interactions lacked biological motivation. I still think this is the case. Why would the effects of helpers depend on egg position or clutch size? What hypothesis is being tested here?

20. There is no explanation as to relevance of egg position, although it is included in all the models. More importantly I don't understand why it is used as a criteria for exclusion of nearly half the available egg data (L540). It varies within a clutch, whereas the variables of interest (helper number) vary across clutches, and so I don't really see why egg position would affect these focal results. So, apart from it being nice to have a model with all explanatory variables in, I don't see the logic of MASSIVELY reducing the power of the analysis on behalf of this variable. Please present results using data for all egg data, to show that the pattern still holds. This would also in part address the issue of how 'demanding' bivariate models might be with this data.

21. L 567 I suspect clutch size is very underdispersed with respect to a Poisson (it is notoriously so), I am surprised that this would fit well - simulating data Poisson with a mean of 2 results in a reasonable proportion of 4+ and that is with no additional variation between clutches/individual. Underdispersion would act to increase the false negative rate in these models. 

22. L 590 Why was breading season not included as random term in the analysis of number of clutches?

23. It doesn't appear that maternal age is considered in the any analysis. Female reproductive output is known to change with age in many systems, and also conceivable that group size may also change with a dominant females age. Could age be driving any of the reported effects?

Joel Pick

---

## [Decision Letter · Decision Letter 2]

1 Aug 2023

Dear Dr Capilla-Lasheras,

Thank you for your patience while we considered your revised manuscript "Mothers front-load their investment to the egg stage when helped in a wild cooperative bird" for consideration as a Research Article at PLOS Biology. Your revised study has now been evaluated by the PLOS Biology editors, the Academic Editor and one of the original reviewers (reviewer #4).

IMPORTANT: You'll see that reviewer #4 still harbours some concerns. I discussed these with the Academic Editor, who sent me the following advice (somewhat edited), which I hope you will find useful:

"The authors have worked hard to deal with extensive reviewer comments and seem to have been diligent. However, I do suggest that you ask the authors to amend the paper in the light of the last set of comments of reviewer #4. I would make it clear that they don't have to make all the changes he required, but, in cases where they do not, they should put in writing why. This way there is a clear written record and it is possible to adjudicate as to the validity of their responses... The one change they should definitely make, however, is to include all the models they ran in the Supp Info. I am less certain about the full vs half dataset issues. In my experience authors usually have a good reason to exclude data. I would give the authors the benefit of the doubt here - but, again, ask them to be precise and put in writing - in the Supp Info or main text - exactly why they did not use the 'full dataset' or think it is not really suitable to draw the strongest conclusions."

In light of the reviews, which you will find at the end of this email, we are pleased to offer you the opportunity to address the [comments/remaining points] from the reviewers in a revision that we anticipate should not take you very long. We will then assess your revised manuscript and your response to the reviewers' comments with our Academic Editor aiming to avoid further rounds of peer-review, although might need to consult with the reviewers, depending on the nature of the revisions.

**IMPORTANT - SUBMITTING YOUR REVISION**

*Resubmission Checklist*

*Published Peer Review*

*PLOS Data Policy*

*Blot and Gel Data Policy*

Sincerely,

Roli Roberts

Roland Roberts, PhD

Senior Editor

PLOS Biology

rroberts@plos.org

REVIEWERS' COMMENTS:

Reviewer #4: 

[identifies himself as Joel Pick]

The comments from my last review have generally been addressed. However, from what is presented between supplements and the response to reviewers, the results seem quite dependent on the model specification, and on the exact dataset used. 

My purpose in suggesting some of these additional analyses is for them to act as sensitivity analyses. Statistical analysis is complex, and there are often many defensible ways to model something. Although I feel that some of the suggestions I have made have a clear root in statistical literature (e.g. the use of bivariate models versus within subject centering), if they give a similar answer than it is up to the authors to choose and defend their particular choice. These sensitivity analyses should then be clearly reported, and act to validate the findings presented. In this case, although many of the additional analyses suggested have been done, they are either not presented in the main text or supplements (just in the response to reviewers), or are presented in the supplements but somewhat disingenuously referred to in the main text (e.g. l 611-614). These additional analyses generally show results in the same direction, but in all cases the ones retained in the main text show a 'significant' result and the others non-significant effects. Now I am not particularly bothered by statistical significance at an arbitrary threshold, but this is the paradigm that is used in this manuscript to assess presence or absence of effects, and so this pattern of results being just below p=0.05 in the main text and above p=0.05 in sensitivity analyses does concern me. In some cases there is also a considerable shift in effect sizes between analyses (for example, when the 'full' dataset is used in the egg size analysis), but the same conclusions are presented. 

Follow up on previous comments:

1. Use of the 'full' dataset.

The change in effect size between the models run on different datasets is considerable. The effect size halves when the full dataset is used, and essentially becomes the same as the effect size seen in males in the reduced dataset (0.01 in females in full dataset versus 0.008 in males in reduced dataset(l 225)). However, this effect in females seems to be considered meaningful (l 611 - 'we also verified that our main conclusions are confirmed when the expanded dataset is analysed'), whereas this non-significant effect in males is described as there being no effect (e.g. l 309 'sparrow-weaver mothers appear to significantly adjust egg size according to female helper number and not male helper number').

In the response to reviewers, it is stated that this additional data is not suitable because clutch size is not known with high certainty. First, clutch size appears to have no affect on egg size, and is not affected by helper number, and there is no interaction effect on egg size between the two in the 'reduced' dataset. Second, it should be clear how often eggs are lost in this reduced dataset, and so what kind of uncertainty this would produce (although if clutch size doesn't correlate with helper number or egg size, it is not clear how important this uncertainty would be). Third, not finding a clutch on the first day of laying (and so the potential for missed eggs) doesn't seem to lead to the exclusion of nests from the analysis of clutch size (l 667).

I am also not convinced of how important it is to know the egg position for the analysis in question. Unlike clutch size, egg position does have a clear 'main' effect on egg size. However, it is not clear how this effect, when measured at the level of the observation (i.e. within clutches) should impact the within and between *mother* effects of helper number. As clutch size is not affected by helper number, there seems no reason why egg position should systematically vary with help number. I think dropping the egg position effect from the reduced dataset analysis would give some idea of any confounding effect - if the same change in effect size of the within mother effect of helper number is seen, this would indicate that there was something confounding here. If the result does not change, then I do not think there is a clear reason why the reduced dataset findings should be considered more reliable than a dataset double the size? A very quick look at this using the data provided on dryad shows that excluding egg position from this analysis does not change the within mother effect of helper number.

Also based on the description in the supplements, some aspects of egg position would be known for the additional nests, for example in nests with three eggs, when two were discovered on the same day, it would be known when the other egg was laid. There are several ways of including missing data in analyses, and here it seems that data on over half the eggs would be known, and so these methods would be feasible. It is up to the authors to decide how to subset their data, but I am not convinced by the statistical justifications given to half the dataset.

Finally, please present standard errors in the SM, not just lone effect sizes (and full model results would be preferable).

2. Bivariate models.

Please provide the full results of the bivariate analyses in the supplements. In the responses to reviewers the correlations have been calculated and compared. As per the references in the last reviews I gave, the important thing to compare here are the *slopes* (covariance/variance in helper number at the same level). These slopes will also be directly comparable to the slopes estimated in the within/among subject centering approach. As such, it is difficult to compare the two approaches, and the comparison between within and among mother effects are not meaningful. As noted in the beginning of my review, the credible intervals of the correlation estimates overlap 0 - presenting the derived slopes and their credible intervals would help interpret how large that overlap is in these analyses.

3. The relationship between egg size and provisioning rate.

Thank you for adding this analysis. It is worth noting that there can be missing data in one or both of the responses in these models in MCMCglmm (it can just be included as NAs), and so the entire provisioning and egg size datasets can be used, even if they don't fully overlap. This will provide greater confidence to the between mother effects, and so may help estimate the correlations with more precision. I would also suggest changing the title in the supplements;'Evidence for a trade-off between egg volume and maternal provisioning rate within mothers' is quite misleading as no trade-off between the two is detected. Also please present the full results of the model in a table.

Other comments:

l 234 - please present full model results in the supplements.

l 614 'and could be substantially larger in some contexts than that reported in in the main paper).'

Or substantially smaller!! On average they are smaller, and >70% nests (based on provided data) have a clutch size of 2, in which there is no within female effect in this extended analysis.

l 679 'Breeding season was not included as a random intercept in this analysis as it perfectly covaries with rainfall (i.e., all mothers were exposed to the same amount of rainfall per breeding season).'

I do not understand this argument, and I don' think it has any statistical basis. Any non-independence in the dataset needs to be modelled, otherwise there is a risk of underestimating standard errors. There are many parts of this analysis where a covariate is included that varies at the same level as a random effect. For example random effects of clutch ID are modelled in the same model that includes clutch size. Indeed, breeding season needs to be included in part because you are modelling something that varies at that level - otherwise you may appear to be over confident (reduced standard errors) in the estimates of rainfall. Perhaps what is meant is that rainfall explains all the among breeding season variation in the number of clutches? If so I think it would be clearer to describe in a different way, and I still don't see why breeding season wouldn't be included, even if it explained 0 variance.

Tables 1-4 It would be worth keeping the order of male and female helper effects consistent

l301 in SM - Should it be 'bivariate' instead of 'binomial'?

Generally, full results of models for all analyses in the main text and supplements need to be presented in the supplements.

Joel Pick

---

## [Editor Report · Decision Letter 3]

15 Sep 2023

Dear Pablo,

Thank you for your patience while we considered your revised manuscript "Mothers front-load their investment to the egg stage when helped in a wild cooperative bird" for publication as a Research Article at PLOS Biology. This revised version of your manuscript has been evaluated by the PLOS Biology editors and the Academic Editor.

Based on our Academic Editor's assessment of your revision, we are likely to accept this manuscript for publication, provided you satisfactorily address the following data and other policy-related requests.

IMPORTANT - Please attend to the following:

a) Please change the Title to something more explicitly informative for our broader readership. We suggest something like, "A decade-long study of a wild cooperative bird shows that mothers invest more in the pre-natal stages of their offspring when non-breeding helpers assist with post-natal care" - feel free to discuss alternatives with me, if this one feels wrong.

b) Please provide a blurb, according to the instructions in the submission form.

c) Many thanks for providing the very helpful data and code deposition in Github, please now provide a permanent DOI'd version in Zenodo, and cite the location of the data clearly in all relevant main and supplementary Figure legends, e.g. “The data underlying this Figure can be found in https://doi.org/10.5281/zenodo.XXXXX”

We expect to receive your revised manuscript within two weeks. 

*Published Peer Review History*

*Press*

Sincerely,

Roli

Roland Roberts, PhD

Senior Editor,

rroberts@plos.org,

PLOS Biology

DATA NOT SHOWN?

---

## [Editor Report · Decision Letter 4]

29 Sep 2023

Dear Pablo,

Thank you for the submission of your revised Research Article "Mothers in a cooperatively breeding bird increase investment per offspring at the pre-natal stage when they will have more help with post-natal care" for publication in PLOS Biology. On behalf of my colleagues and the Academic Editor, Michael Jennions, I'm pleased to say that we can in principle accept your manuscript for publication, provided you address any remaining formatting and reporting issues. These will be detailed in an email you should receive within 2-3 business days from our colleagues in the journal operations team; no action is required from you until then. Please note that we will not be able to formally accept your manuscript and schedule it for publication until you have completed any requested changes.

Sincerely, 

Roli

Senior Editor

PLOS Biology

rroberts@plos.org